# The corticospinal tract primarily modulates sensory inputs in the mouse lumbar cord

Yunuen Moreno-Lopez[†‡], Charlotte Bichara[†§], Gilles Delbecq, Philippe Isope, Matilde Cordero-Erausquin*

Institut des Neurosciences Cellulaires et Intégrées, CNRS - Université de Strasbourg, Strasbourg, France

**\*For correspondence:**
cordero@unistra.fr

[†]These authors contributed equally to this work

**Present address:** [‡]Burke Neurological Institute, White Plains, New York, United States; [§]Neuro-Electronics Research Flanders (NERF), Leuven, Belgium

**Competing interest:** The authors declare that no competing interests exist.

**Abstract** It is generally assumed that the main function of the corticospinal tract (CST) is to convey motor commands to bulbar or spinal motoneurons. Yet the CST has also been shown to modulate sensory signals at their entry point in the spinal cord through primary afferent depolarization (PAD). By sequentially investigating different routes of corticofugal pathways through electrophysiological recordings and an intersectional viral strategy, we here demonstrate that motor and sensory modulation commands in mice belong to segregated paths within the CST. Sensory modulation is executed exclusively by the CST via a population of lumbar interneurons located in the deep dorsal horn. In contrast, the cortex conveys the motor command via a relay in the upper spinal cord or supraspinal motor centers. At lumbar level, the main role of the CST is thus the modulation of sensory inputs, which is an essential component of the selective tuning of sensory feedback used to ensure well-coordinated and skilled movement.

## Introduction

Several brain and spinal structures have been involved in the generation and control of movement in a task-dependent manner. The primary motor cortex is classically associated to forelimb skilled movements (*Starkey et al., 2005*; *Guo et al., 2015*; *Miri et al., 2017*; *Galiñanes et al., 2018*) and to complex locomotor tasks (obstacles crossing, leaning scale or beam, etc.; *Beloozerova and Sirota, 1993*; *DiGiovanna et al., 2016*; *Wang et al., 2017*; *Bieler et al., 2018*) while its role in rythmic locomotion appears, if anything, minor (*Beloozerova and Sirota, 1993*; *DiGiovanna et al., 2016*). A coordinated movement relies not only on a proper motor command, but also on the online analysis of an ongoing multisensory feedback indicating whether the movement has reached the planned objective or requires correction (*Akay et al., 2014*; *Fink et al., 2014*; *Azim and Seki, 2019*). To optimize the analysis of the overwhelming quantity of reafferent information, specific tuning of peripheral feedback is possible through inhibition of sensory information at different stages of the sensory pathway, a phenomenon known as 'sensory gating' (*Seki and Fetz, 2012*; *McComas, 2016*).

At the spinal cord level, sensory gating can be mediated by presynaptic inhibition of sensory inputs through primary afferent depolarization (PAD), a powerful mechanism controlled both by supraspinal centers and by adjacent primary afferents (*Eccles et al., 1962*). While multiple mechanisms have been demonstrated (*Shreckengost et al., 2021*), the most studied cause of PAD is presynaptic GABA-A-mediated depolarization. Primary afferents have indeed a high chloride content and respond to GABA by a depolarization that has been shown to reduce the amplitude of incoming action potentials and thus inhibit synaptic transmission (*Prescott and De Koninck, 2003*; *Doyon et al., 2011*). PAD has been involved in the adjustment of reflex excitability (*Rudomin and Schmidt, 1999*), and its absence results in a pathological context associated with spasticity (*Caron et al., 2020*). Proprioceptive

feedback is required for skilled locomotion (*Akay et al., 2014*), and presynaptic inhibition of sensory feedback is essential for performing smooth forelimb skilled movements in mice (*Akay et al., 2014*; *Fink et al., 2014*). Interestingly, PAD can also be evoked by stimulation of the sensorimotor cortex in primates, cat, and rats (*Andersen et al., 1962*; *Carpenter et al., 1963*; *Andersen et al., 1964*; *Abdelmoumène et al., 1970*; *Eguibar et al., 1994*; *Wall and Lidierth, 1997*). This cortically evoked PAD may underlie cortical sensory gating at the spinal level and contribute to the cortical involvement in voluntary movements, in addition to the classical drive of motoneurons. In this study, we aimed at elucidating the neuronal pathways conveying the cortical motor command and sensory control to the hindlimb in mice. This cortical control of the spinal cord, either to trigger motoneurons or to modulate sensory information, can rely on multiple pathways, the corticospinal tract (CST) being the most direct one. Its involvement in motor control is long acknowledged, although its precise contribution might vary depending on the motor task (locomotion vs. skilled movement) or targeted limb (forelimb vs. hindlimb) (*Miri et al., 2017*; *Wang et al., 2017*; *Karadimas et al., 2020*). The CST has also been involved in cortically evoked PAD in cats as a pyramidotomy abolishes it (*Rudomin et al., 1986*). However, discriminating sensory vs. motor contribution has been hampered by the lack of proper tools to segregate functionally or anatomically overlapping paths.

Importantly, none of the CST functions can rely exclusively on direct CST contacts onto motoneurons (MN) as these represent the exception rather than the rule. Indeed, monosynaptic CST-MN contacts are present only in some higher primates, while CST-interneuronal connections are preponderant (in adults) across species (rodents and primates) (*Alstermark et al., 2011*; *Ebbesen and Brecht, 2017*; *Ueno et al., 2018*). In the recent years, specific genetic factors have enabled the identification of several interneuronal populations, and of their inputs, including those from the cortex (*Hantman and Jessell, 2010*; *Bourane et al., 2015*; *Abraira et al., 2017*; *Ueno et al., 2018*). But neurons with identical genetic markers do not necessary share identical function: in these genetically distinct populations, up to 65% of the neurons receive CST inputs (*Ueno et al., 2018*), while there is a significant fraction of neurons that is not targeted by the CST. Although manipulating these populations has been extremely informative to better understand their role, it remains unclear whether the changes observed upon these manipulations can be specifically attributed to the neuronal subset that receives inputs from the CST or to the subset that does not. Therefore, it is still difficult to assess the specific role of the targets of the CST in movements.

In this study, we have investigated the corticofugal paths conveying motor command and modulation of sensory inputs through PAD. We have in particular interrogated the contribution of the CST and its direct lumbar targets through the development of an innovative intersectional viral strategy. We show that cortically evoked PAD in lumbar primary afferents is conveyed exclusively by the CST through a population of lumbar interneurons. In contrast, our results suggest that motor command to the hindlimbs is relayed either through supraspinal motor centers, or through the CST with a propriospinal relay in the upper cord. The role of the lumbar CST thus appears to be mainly the modulation of sensory inputs, which may in turn selectively regulate sensory gain involved in the refinement of motor control during movement.

## Results

### PAD and muscular contractions originate in the same cortical area

The rodent sensorimotor cortex has a highly somatotopic organization (*Ayling et al., 2009*). Mainly studied in a motor perspective, it has been repeatedly mapped for its ability to provoke motor contraction (*Li and Waters, 1991*; *Ayling et al., 2009*), but rarely for its ability to induce PAD (*Wall and Lidierth, 1997*). Cortically evoked PAD may inhibit the transmission of sensory information from primary afferents to the central nervous system and can be experimentally assessed by recording dorsal root potentials (DRPs, *Figure 1A*). We thus recorded DRPs in vivo in lumbar roots L4–L6 (conveying hindlimb sensory inputs) of Thy1-ChR2 mice, expressing channelrhodopsin2 in most neuronal cells including layer V cortical neurons (*Arenkiel et al., 2007*). DRP recordings were performed in isoflurane-anesthetized mice, and trains of photostimulations (5 × 8 ms pulses, 1 ms apart) were applied to activate ChR2 at different positions of the surface of the sensorimotor cortex according to a 500-μm-spaced grid. The area inducing the DRPs was centered at AP = −0.75 mm, L = 1.5 mm (*Figure 1C*, n = 7 mice). Similar photostimulation of this area in mice expressing only

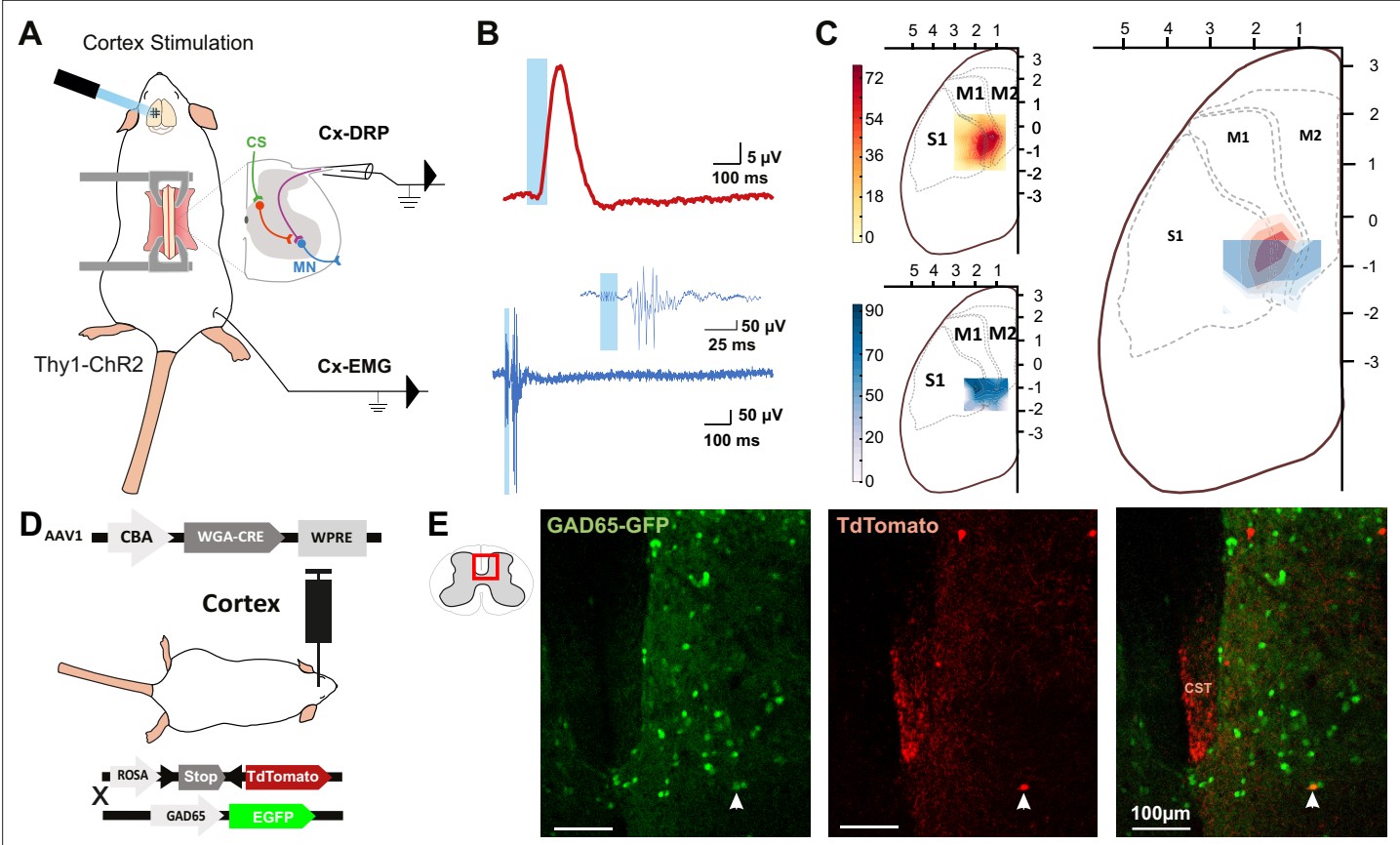

**Figure 1.** The same cortical area evokes dorsal root potentials (DRPs) and muscular contractions of the hindlimbs. (**A**) Schematic experimental design for cortically evoked (Cx-) DRP and electromyographic (EMG) recordings elicited by photostimulation of the contralateral sensorimotor cortex of Thy1-ChR2 mice. DRPs correspond to the presynaptic depolarization of primary afferents propagating antidromically to the suction recording electrode containing a lumbar dorsal root, while EMGs are recorded in the tibialis anterior (TA) muscle. (**B**) Representative traces of a cortically evoked DRP (top red trace, average of 30 sweeps) and EMG (bottom blue trace, average of three sweeps) after photostimulation (blue window) of the contralateral cortex. (**C**) Left: heatmaps of the amplitude of the responses of cortically evoked DRPs (top) and EMGs (bottom) in % or the maximum response. Right: overlap of the two maps (red: DRP; blue: EMG) presented as isopotential contour plots (three color grades corresponding to 37, 50, and 63% of maximum value). Coordinates of the cortex are expressed in mm and centered on *Bregma*. M1: primary motor cortex; M2: secondary motor cortex; S1: primary sensory cortex, according to the Paxinos atlas (***Kirkcaldie et al., 2012***). (**D**) Schematic experimental design to identify GAD65-expressing neurons amongst corticospinal neurons targets: AAV2/1-CBA-WGA-CRE was injected in the sensorimotor cortex of TdTomato-flex X GAD65-GFP mice. Analysis of extend of the injection site is presented in ***Figure 1—figure supplement 1*** and of the monosynaptic nature of the transsynaptic tracing in ***Figure 1—figure supplement 2***. (**E**) Photomicrographs (z-projection of confocal images) of a GAD65-GFP mouse lumbar dorsal horn (localization of the view indicated in the inset) after transsynaptic labeling from the hindlimb sensorimotor contralateral cortex. A target of the CST (expressing TdTomato after transsynaptic transfer of WGA-Cre) also expresses GAD65-GFP. CST: corticospinal tract. Similar experiment in ChAT-EGFP mice is presented in ***Figure 1—figure supplement 3***.

The online version of this article includes the following figure supplement(s) for figure 1:

**Figure supplement 1.** Sensorimotor injection site analyses: cortical layer V fluorescence quantification.

**Figure supplement 2.** Monosynaptic transsynaptic tracing from the cortex.

**Figure supplement 3.** Postsynaptic corticospinal neurons do not colocalize with ChAT.

the fluorescent reporter EGFP in corticospinal (CS) neurons leads to no DRP signal (see below and ***Figure 3—figure supplement 1***). On a different group of animals, electromyographic (EMG) recordings were performed in the tibialis anterior (TA) under ketamine/xylazine anesthesia because EMGs cannot be evoked from cortical stimulation under isoflurane anesthesia (see Materials and methods). The sensorimotor cortex was similarly mapped with trains of photostimulations (6 × 1 ms pulses, 2 ms apart) and the area inducing EMG was centered at AP = −0.75 mm, L = 1.75 mm (n = 5 mice; ***Figure 1C***). Again, similar photostimulation of this area in mice expressing only EGFP in CS neurons

leads to no EMG signal (see below and *Figure 3—figure supplement 1*). We observed that the two zones are concentric, demonstrating that cortically evoked PAD and motor command of the hindlimb can originate from the same cortical area.

## Spinal targets of the CST

We next investigated the spinal circuit underlying cortically evoked DRPs. DRPs can be segmentally evoked by the activation of a neighboring root, dampening sensory inputs predominantly through GABA-dependent primary-afferent depolarization (*Rudomin and Schmidt, 1999*). GABAergic terminals presynaptic to proprioceptive fibers arise from GAD65 interneurons (*Hughes et al., 2005*; *Betley et al., 2009*) and their activation produces DRPs (*Fink et al., 2014*). We investigated whether the CST directly targets GAD65-expressing spinal lumbar neurons that could in turn lead to inhibition of primary afferents. In TdTomato-flex mice crossed with GAD65-GFP mice, we injected a monosynaptic anterograde transsynaptic virus (AAV2/1-CBA-WGA-Cre) encoding the Cre-recombinase fused to the wheat germ agglutinin (WGA; *Libbrecht et al., 2017*) in the area inducing hindlimb DRPs and EMG (*Figure 1—figure supplement 1*). In contrast to its retrograde and anterograde transport when directly injected (*LeVay and Voigt, 1990*), WGA is reported to provide an exclusively anterograde transneuronal tracing when expressed in transgenic mice (*Braz et al., 2002*) or virally, including in the form of a WGA-Cre fusion (*Gradinaru et al., 2010*; *Libbrecht et al., 2017*), but see *Xu and Südhof, 2013*. We indeed observed that the fusion with WGA provided transsynaptic properties to the Cre recombinase, which was transferred to the cortex's monosynaptic targets where it triggered the expression of TdTomato (see Materials and methods, *Figure 1—figure supplement 2*). While a retrograde transsynaptic labeling is difficult to exclude in the brain due to the reciprocal connections of the cortex with its targets, the absence of direct spino-cortical projection (and absence of labeling in the ascending tracts, see *Figure 1—figure supplement 2E*) ensures that spinal Td-Tomato-expressing neurons are direct targets of the CST. We evaluated the proportion of GAD65+ neurons among these neurons (*Figure 1D,E*): 16.4% ± 3.2% of spinal Td-Tomato-positive neurons were GAD65-GFP+ (69 neurons out of 427, n = 3 mice), suggesting that the cortically evoked DRP may be mediated directly by the activation of a GAD65+ spinal target of the CST. In the spinal dorsal horn, cholinergic neurons are a subpopulation of GABAergic neurons (*Todd, 1991*; *Mesnage et al., 2011*), and cholinergic terminals are known to be presynaptic to primary afferents (*Ribeiro-da-Silva and Cuello, 1990*; *Pawlowski et al., 2013*) and have recently been shown to modulate primary afferent inputs (*Hochman et al., 2010*; *Shreckengost et al., 2021*). By performing a similar experiment in ChAT-EGFP mice, we showed that only 2.2% ± 1.8% of CST targets were ChAT-EGFP positive (1 out of 45 neurons, n = 4 mice), limiting the likelihood of a direct major involvement of a cholinergic mechanism (*Figure 1—figure supplement 3*).

## Segregation of pathways for cortically evoked DRPs and EMGs

Although lumbar DRPs and tibialis-EMGs are elicited by the same area in the mouse sensorimotor cortex, whether they share the same corticofugal pathway and spinal circuits remains unknown. We first interrogated the contribution of indirect cortical-to-spinal pathways to these two functions (*Figure 2*). Since photostimulation of the sensorimotor cortex in Thy1-ChR2 activates a heterogeneous population of layer V cortical neurons (*Arenkiel et al., 2007*), we selectively lesioned the direct CST (*Figure 2B*) using an acute electrolytic lesion at the level of the pyramidal decussation (pyramidotomy) (*Figure 2—figure supplement 1A, C*). In this study, we have consistently performed a caudal pyramidotomy, sparing CST collaterals targeting brainstem motor nuclei that branch more rostrally (*Akintunde and Buxton, 1992*; *Alstermark and Pettersson, 2014*) as well as rubrospinal or reticulospinal tracts that can be activated by corticorubral or corticoreticular neurons (*Figure 2B*). Cortically evoked DRPs were completely abolished by this pyramidotomy (DRP amplitude before: 3.8 µV ± 0.32 µV; after: 0.30 µV ± 0.12 µV, n = 3 mice, p=0.006), whereas cortically evoked EMGs were hardly affected (S/N ratio 3.3 ± 0.61 before, 2.2 ± 0.1 after, n = 4 mice, p=0.517, all Z-scores above significance after lesion, *Figure 2—figure supplement 2B*; *Figure 2C,D*). This demonstrates that indirect cortical-to-spinal pathways (involving supraspinal motor centers) do not encode cortically evoked DRPs but have a major role for cortically evoked motor contraction of the hindlimb. These results show that, in the mouse lumbar cord, the CST mediates cortically evoked DRPs and hence modulates sensory inputs in a presynaptic manner.

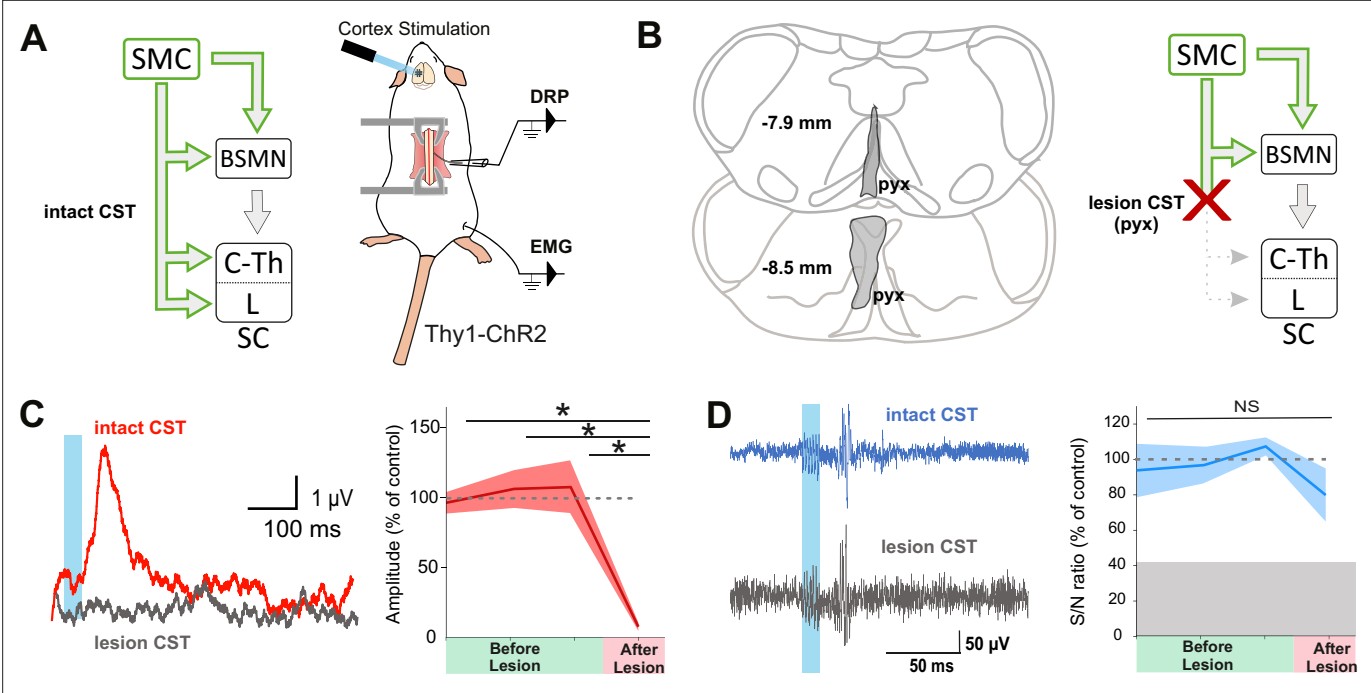

**Figure 2.** The corticospinal tract (CST) is essential for cortically evoked inhibition of primary afferents. (**A**) Left: diagram showing direct and indirect cortical-to-spinal paths. In green: paths potentially activated by photostimulation in the cortex of Thy1-ChR2 mice. Right: experimental design. (**B**) Left: drawing showing the extent of the pyramidal lesion (in gray, pyx) in the frontal plane (distance to *Bregma* indicated) in two animals whose recordings are respectively illustrated in (**C**) and (**D**). Right: diagram showing the indirect cortical-to-spinal paths spared by the pyramidotomy. Systematic histological analysis of the pyramidotomies is presented in *Figure 2—figure supplement 1*. (**C**) Left: representative traces of cortically evoked dorsal root potentials (DRPs) recording before (red) and after (gray) CST lesion (average of 30 traces). Right: the pyramidotomy abolishes cortically evoked DRPs (p=0.006, one-way ANOVA, n = 3 mice). Post-hoc test Holm–Sidak, *p<0.05 (**D**) Left: representative traces of cortically evoked electromyographic (EMG) recording before (blue) and after (gray) pyramidal lesion (average of three traces). Right: there is no significant change in the S/N ratio of the EMG response before and after the pyramidal lesion (93,8, 96,9, and 109,3%  prior to pyramidotomy to 77.2%  after, p=0.517, one-way ANOVA, n = 4 mice). Gray zone: noise level (see Materials and methods). EMG Z-scores for individual mice before and after pyramidotomy are presented in *Figure 2—figure supplement 2*.

The online version of this article includes the following figure supplement(s) for figure 2:

**Figure supplement 1.** Histology of the pyramidotomies.

**Figure supplement 2.** Z-score of individual electromyographic (EMG) responses assessing the significance of the corresponding S/N ratio.

Although the contribution of the rubrospinal or reticulospinal tracts to motor command is well acknowledged in both rodents and primates, the CST is also believed to encode motor command (*Wang et al., 2017*). This could have been underrated by simultaneous stimulation of the indirect cortical-to-spinal pathways in Thy1-ChR2 mice. We thus interrogated the specific contribution of the CST to motor contraction by targeting exclusively these neurons through injection of a ChR2-encoding retrograde virus (AAVrg-CAG-hChR2-H134R-tdTomato) in the lumbar spinal cord (*Figure 3A*). As expected, the infected CS neurons were located in the area delimited by the previous functional mapping of hindlimb muscle contraction and DRP (*Figure 3B,C*). Photostimulation of CS neurons (at the cortical level) induced a robust EMG signal (S/N ratio 2.4 ± 0.52, n = 5 mice, *Figure 3E*, upper), demonstrating that CS neurons evoke motor contraction. In agreement with the pyramidotomy results (*Figure 2C*), the photostimulation of CS neurons also induced DRPs (mean amplitude = 14.5 ± 4.0μV, n = 5), while photostimulation in control mice expressing GFP in CS neurons induced neither an EMG (all Z-scores below significance) nor a DRP signal (mean amplitude = 0.3 ± 0.0 μV, n = 4, *Figure 3—figure supplement 1*). Because some CS neurons send collaterals to supraspinal regions involved in motor command (*Akintunde and Buxton, 1992*), we next addressed whether the motor command traveled through the direct cortical-to-spinal branch of CS neurons or via supraspinal relays. A pyramidotomy, similar to the ones performed above (*Figure 2—figure supplement 1C*), completely abolished the EMG signal evoked by cortically activating CS neurons (S/N ratio 0.9 ± 0.12; n = 3 mice, p=0.013; all Z-scores below significance after pyramidotomy, *Figure 2—figure supplement 2C*; *Figure 3D,E*).

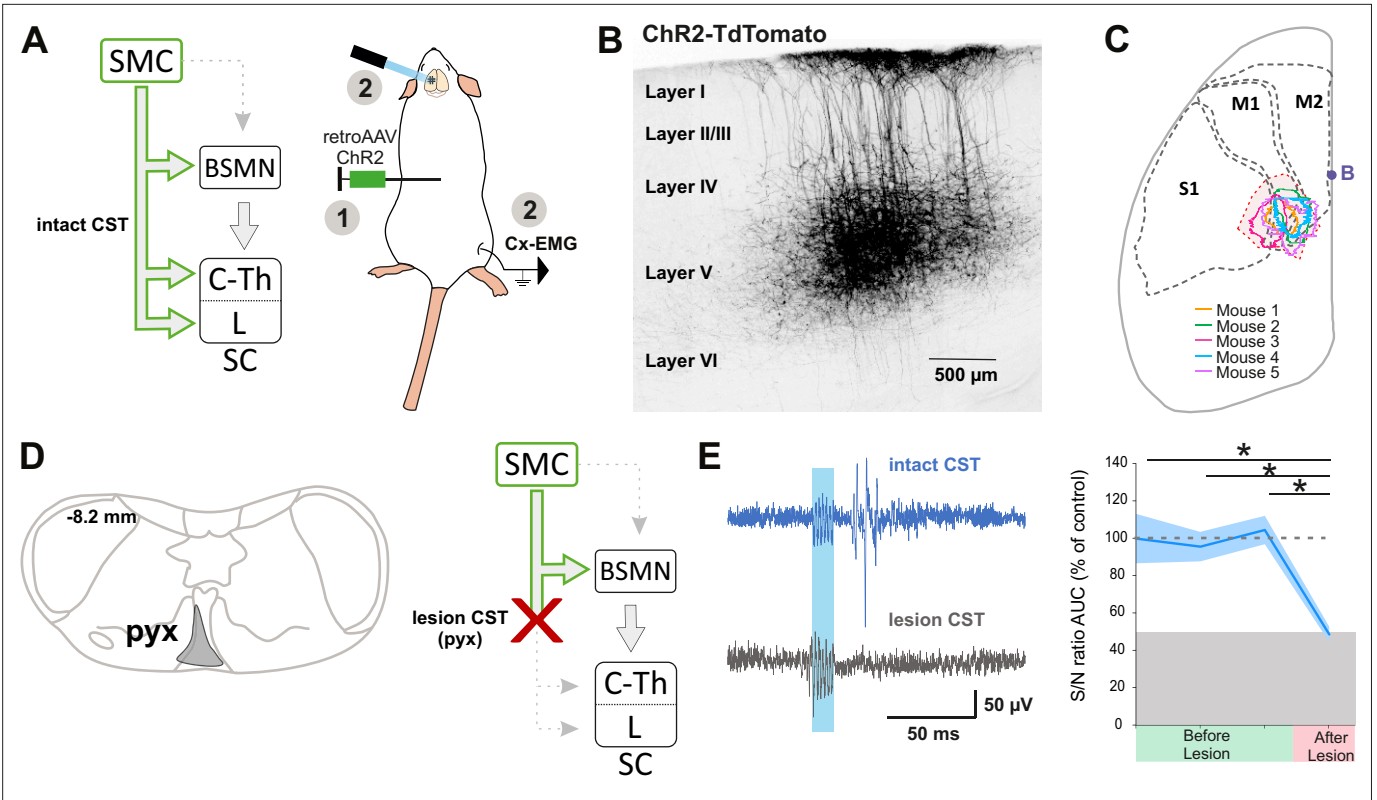

**Figure 3.** The corticospinal (CS) tract can encode muscular contractions. (**A**) Left: diagram illustrating CS neurons and their collaterals expressing ChR2 after retrograde infection in the lumbar cord with a ChR2-retro AAV. Right: experimental design: the photostimulation and electromyographic (EMG) recording session took place at least 3 weeks after the infection. (**B**) Sagittal image from the sensorimotor cortex (lateral = 0.68 mm), showing retrogradely labeled CS neurons (expressing ChR2-TdTomato). (**C**) Drawing of a cortex top view, showing the localization of retrogradely labeled CS neurons in the five animals (solid lines). They are within the area functionally determined as inducing dorsal root potentials (DRPs) and EMGs in the hindlimb (dashed lines). (**D**) Left: drawing showing the extent of the pyramidal lesion in the animal whose recordings are illustrated in (**E**). Right: diagram showing the supraspinal collaterals of ChR2-expressing CS neurons spared by the pyramidotomy. Systematic histological analysis of the pyramidotomies is presented in *Figure 2—figure supplement 1*. (**E**) Left: cortically evoked EMG recordings before (blue) and after (gray) the pyramidal lesion. Right: the pyramidotomy abolishes cortically evoked EMG (p=0.012, one-way ANOVA, n = 5 mice). Post-hoc test Holm–Sidak, *p<0.05. Gray zone: noise level (see Materials and methods). EMG Z-scores for individual mice before and after pyramidotomy are presented in *Figure 2—figure supplement 2*. The blue window on recording traces indicates photostimulation. SMC: sensorimotor cortex; BSMN: brain stem motor nuclei; C-Th: cervico-thoracic spinal cord; L-SC: lumbo-sacral spinal cord.

The online version of this article includes the following figure supplement(s) for figure 3:

**Figure supplement 1.** Dorsal root potentials (DRPs) are induced by photostimulation of ChR2-expressing corticospinal (CS) neurons (but not by GFP-expressing CS neurons).

This demonstrates that, while supraspinal collaterals of the CST do not trigger motor output, direct spinal branches can drive motor command. However, this contribution does not seem significant when other cortical neurons are co-activated (*Figure 2C*).

## Spinal targets of the CST evoke DRPs but not muscular contraction

Because both DRPs and EMGs of a given limb are encoded by the same sensorimotor cortical area and can be conveyed by the spinal branch of the CST, we next interrogated whether these functions are segregated at the spinal level. CS neurons projecting to the lumbar spinal cord can give off collaterals at the cervico-thoracic level (*Kamiyama et al., 2015*; *Karadimas et al., 2020*) and contact cervical propriospinal neurons (*Ueno et al., 2012*; *Ni et al., 2014*) that in turn project to lumbar moto-neurons to relay motor command (*Ni et al., 2014*). Indeed, many experiments demonstrated that these neurons are involved in segmental coordination (*Miller et al., 1975*; *Nathan et al., 1996*; *Reed et al., 2006*; *Reed et al., 2009*). In order to test the contribution of the lumbar branch of CS neurons and rule out antidromic stimulation of rostral collaterals, we combined the transsynaptic tool used in

*Figure 1* with an intersectional approach at the lumbar level. We first identified the lumbar segments containing CST targets by using cortical injection of AAV2/1-CBA-WGA-Cre in TdTomato-flex mice (*Figure 4A*): CST targets were concentrated in spinal lumbar segments L2–L3, rostral to the large pools of motoneurons. They were largely located in the dorsal horn (85.4% were dorsal to the central canal), in particular in the contralateral deep dorsal horn (laminae IV to VII) (*Figure 4B*, *Figure 4—figure supplement 1*). Consistently with the monosynaptic nature of the transynaptic labeling, their number was not correlated with the delay between the infection and the histological analysis (*Figure 4—figure supplement 1*). CST targets were mostly located in the medial zone, lateral to the dorsal funiculus, where CST fibers penetrate the gray matter. We thus combined the cortical injection of AAV2/1-CBA-WGA-Cre with the spinal (L4) injection of an AAV (AAV9-Ef1a-DIO-ChETA-EYFP) encoding for a Cre-dependent form of ChETA (a ChR2 variant *Gunaydin et al., 2010*; *Figure 4C*). ChETA-EYFP expression was therefore restricted to the lumbar targets of the CST (*Figure 4C and D*). Our transectional approach induced ChETA-EYFP expression even in the deepest targets of the CST in the ventral horn (*Figure 4—figure supplement 2*). There was no ChETA-EYFP labeling in the dorsal funiculus, demonstrating that the intraspinal AAV injection did not lead to retrograde infection of CS neurons (*Figure 4—figure supplement 2*). Intraspinal injections did not affect the general behavior of mice (e.g., no visible motor deficits nor pain-like behaviors) nor induced long-lasting weight loss. As surface photostimulation of the spinal cord in Thy1-ChR2 animals produced an EMG signal and directly activated neurons even in the deep dorsal horn (*Figure 4—figure supplement 3*), we performed an identical photostimulation in the animals expressing ChETA in the targets of the CST. This stimulation induced a spinal LFP observable as deep as 1150 μm from the surface (*Figure 4—figure supplement 3*). Although the number of ChETA-expressing neurons was highly variable (due to the variability inherent to the WGA-Cre transynaptic tool, see *Figure 4—figure supplement 1*, and to the intersectional strategy), spinal photostimulation induced DRPs in the ipsilateral lumbar root of eight out of nine mice (2.93 ± 0.48 μV, n = 8 mice, *Figure 4E–G*), demonstrating that DRPs can be directly controlled by the CST projecting to the lumbar cord through local interneurons. Interestingly, the amplitude of the DRPs was not related to the number of ChETA-expressing spinal neurons, suggesting that probably only a fraction of those are involved in primary afferent control. However, photostimulation of these same lumbar targets of the CST (same animals) failed to elicit hindlimb movements or muscle contraction as attested by EMG recordings (S/N ratio 1.0 ± 0.02; n = 7 mice, Z-scores below significance, *Figure 2—figure supplement 2D*; *Figure 4E–G*). Together with the results presented in *Figure 3*, we can conclude that CS neurons are able to evoke motor contraction by activation of their spinal targets, but that these targets are not located in the lumbar cord but rather at another spinal level. Cortically evoked DRPs and EMGs thus follow a similar corticofugal pathway that segregates at the spinal level.

## Discussion

In this study, in order to gain knowledge on the contribution of the CST to the cortical control of movements, we have evaluated its ability to convey the muscle contraction command or to modulate sensory inputs. We found that the modulation of sensory inputs from the hindlimb (DRP in lumbar sensory roots) is exclusively carried out by the CST via a population of lumbar interneurons located in the deep dorsal horn. In contrast, the cortex induces muscle contraction of the hindlimb (TA) through a relay in the upper spinal cord or supraspinal motor centers. The two mechanisms are therefore segregated, and the main role of the lumbar CST appears to be the modulation of sensory inputs.

### Reliability of circuit investigations

We have identified the cortical area involved in these two mechanisms through functional means: cortically evoked DRPs and EMGs. For ethical and technical reasons, most of these recordings were performed on different animals and conditions. First, EMG recordings after cortical stimulation required a light anesthesia level (*Tennant et al., 2011*), only possible using ketamine/xylazine. Such a light anesthesia is not ethically compatible with the large and invasive surgery required for recording cortically evoked DRPs. DRPs were thus recorded under isoflurane anesthesia under which no cortically evoked EMG could be recorded. We therefore acknowledged that the cortical zone responsible for DRPs, smaller than the one responsible for EMGs, might have been underestimated by the

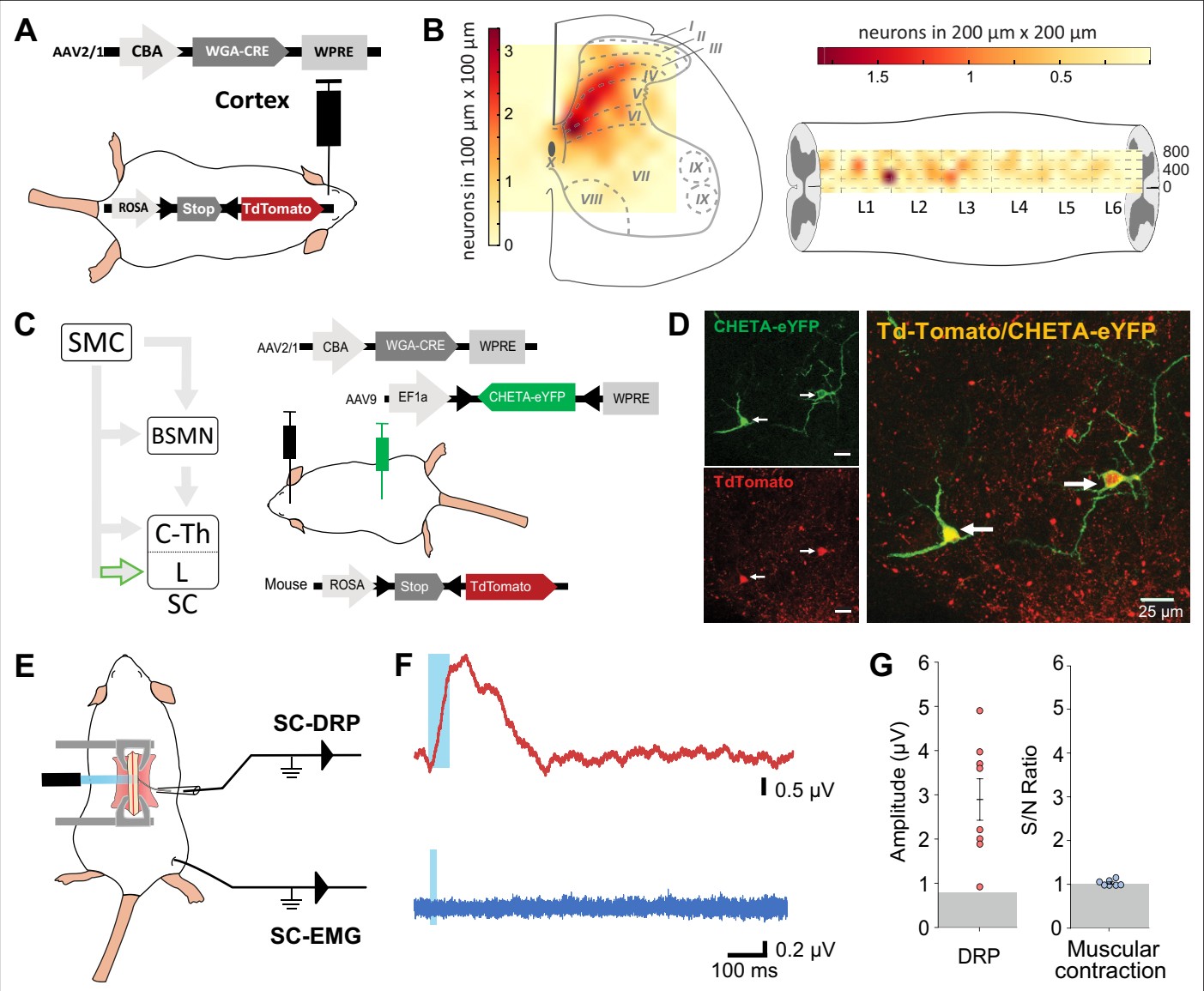

**Figure 4.** Lumbar corticospinal postsynaptic neurons encode dorsal root potentials (DRPs) but no movements. (**A**) Experimental design: the targets of the corticospinal tract (CST) are labeled through a transynaptic approach consisting of AAV2/1-CBA-WGA-CRE injection in the hindlimb sensorimotor cortex of TdTomato-flex mice. (**B**) Localization of the spinal targets of the CST: heatmap showing the distribution of the neurons in the lumbar cord (6 mm long) projected into the transverse plane (average of nine mice; left) or horizontal plane (average of six mice; right) plane. (**C**) Left: diagram illustrating that only lumbar direct targets of the CST express CHETA in the following experiment. Right: experimental design: TdTomato-flex mice received an injection of AAV2/1-CBA-WGA-CRE in the hindlimb sensorimotor cortex and an injection of AAV9-Efl-flex-CHETA-eYFP in the L4 spinal segment. (**D**) Photomicrographs (z projection of confocal images) from the dorsal horn of the spinal cord (laminae V/VI); the arrows point at two targets of the CST expressing TdTomato+ and CHETA-eYFP. (**E**) Experimental design illustrating spinal photostimulation of the lumbar targets of the CST. (**F**) Representative traces of DRP (red trace, average of 60 traces) and electromyographic (EMG; blue trace, average of three traces) recordings from the same animal after photostimulation of the lumbar targets of the CST (blue window). (**G**) Photostimulation of the lumbar targets of the CST induces DRPs (left) but no EMG signal (right). Gray zone: noise level (see Materials and methods). EMG Z-scores for individual mice are presented in *Figure 2—figure supplement 2*. SMC: sensorimotor cortex; BSMN: brain stem motor nuclei; C-Th: cervico-thoracic spinal cord; L-SC: lumbo-sacral spinal cord.

The online version of this article includes the following figure supplement(s) for figure 4:

**Figure supplement 1.** Strategy to label the spinal targets of corticospinal (CS) neurons.

**Figure supplement 2.** Strategy to stimulate exclusively the spinal targets of corticospinal (CS) neurons.

**Figure supplement 3.** Efficacy of the surface spinal photostimulation.

deeper anesthesia. We thus did not attempt to quantitatively compare EMG vs. DRP map extents, but concluded that they partially overlap allowing us to stimulate at the center of two maps. Although the different experimental approaches might also have differentially affected the recruitment threshold for EMG vs. DRP, we do not compare these thresholds in between stimulation/recording configurations. Rather, we analyze the amplitude of a given signal (EMG or DRP) with a given experimental approach in different animal models (THY1, ChR2-retro, pyramidotomy, transynaptically labeled CS targets).

Circuit investigation also relies on the efficiency of pyramidotomies that affect the spinal branch of CS tract while sparing CST collaterals targeting brainstem motor nuclei that are branching more rostrally (*Akintunde and Buxton, 1992*). We performed similar pyramidotomies in two different group of mice, either Thy1-ChR2 or ChR2-retro group (*Figures 1 and 2*, *Figure 1—figure supplement 3*). In the latter group, in which the CST expressed ChR2, we observed a complete loss of cortically evoked EMGs after pyramidotomy. These results confirm that the pyramidotomy was efficient to block CST-induced EMG. Therefore, the remaining EMG signal after pyramidotomy in Thy1-ChR2 mice (*Figure 2D*) is likely the result of a non-CST pathway.

Finally, an intriguing result we report is the absence of EMG signal after photostimulating the targets of the CST at the lumbar level (*Figure 2F,G*). The reliability of this result relies on multiple technical controls. First, the lack of muscle contraction cannot be attributed to damages of spinal microcircuits due to intraspinal AAV injection. None of the mice presented obvious alteration in the behavior nor a long-lasting weight loss. In addition, similar intraspinal injections were performed for the ChR2-retrograde group, and in these animals, EMG was systematically observed after cortical stimulation, demonstrating the integrity of required spinal circuits. Second, the intraspinal AAV injection efficiently infected neurons in the dorsal horn, and even in the dorsal part of the ventral horn, that is, in an area comprising 97% of the lumbar targets of the CST. Third, the optic fiber positioned at the surface of the cord evoked an intraspinal LFP as deep as 1150 μm in the double-infected mice, but also evoked EGMs in Thy1-ChR2 mice, as previously reported (*Caggiano et al., 2016*). We also demonstrate direct photoactivation of neurons located up to 700 μm from the surface with the short 1 ms stimulus used for EMG recordings. As 73% of the targets of the CST are located within 700 μm from the surface, this demonstrates that the large majority of this population can be directly activated by this procedure. Finally, using a similar surface photostimulation with powers equal to or lower than in the present study, Caggiano and colleagues were able to induce motor contraction in ChAT-ChR2 animals and interpreted this as direct activation of motoneurons located in the deep ventral horn (*Caggiano et al., 2016*). We thus consider very likely that the surface illumination we used could reach most of the lumbar CS-postsynaptic neurons, and that the absence of EMG in this situation is biologically relevant.

## Anatomical definition of the hindlimb sensorimotor cortex

In our study, the cortical area inducing hindlimb contraction is similar to that previously reported in mice (*Li and Waters, 1991*; *Ayling et al., 2009*). This cortical area (centered around 1 mm caudal and 1.5 mm lateral to Bregma) spans over the motor cortex and also partly over the primary somatosensory cortex according to some atlases and studies (*Kirkcaldie et al., 2012*; *Liu et al., 2018*; *Karadimas et al., 2020*). We thus use the conservative term 'hindlimb sensorimotor cortex.' Interestingly, we found that stimulation of this area also induced PAD at the lumbar level. This overlap might be a specificity of the 'hindlimb area' of the cortex (HLA) and may not generalize to the 'forelimb areas' (RFA and CFA). Indeed, the CS neurons from the forelimb areas are spatially segregated (forming clusters projecting either to the dorsal or to the intermediate-ventral cervical cord), while those from the HLA are largely intermingled (*Olivares-Moreno et al., 2017*). In addition, our results demonstrate that the same cortical area is labeled after retrograde tracing from the lumbar cord (*Figure 3C*) excluding the existence of a different pool of cortico-lumbar neurons.

## Pathways conveying the cortical motor command

Our study concludes that a large fraction of the cortical motor command to the hindlimb is conveyed through non-CST pathways. These could involve the reticular formation or the red nucleus that receive cortical innervation, project directly or indirectly onto spinal motoneurons, and control different parameters of movements (*Lavoie and Drew, 2002*; *Zelenin et al., 2010*; *Esposito et al., 2014*). Our recordings also suggest that the cortex conveys a motor command to the hindlimb through the

CST projecting to a non-lumbar spinal segment as direct stimulation of the CS lumbar targets failed to produce muscle contraction. We have discussed above possible technical caveats and concluded that our approach most likely induced activation of the vast majority of the lumbar CS targets labeled by the transynaptic tracing. We cannot rule out that some of these lumbar targets contribute to convey the cortical motor command to the hindlimb but were not labeled in large enough quantities through our transynaptic approach. The efficiency of WGA-Cre transynaptic tracing might be target specific (*Libbrecht et al., 2017*), and the number of labeled neurons in the spinal cord suggests that not all targets were labeled. However, this would still implicate that these neurons are distinct from the ones involved in the generation of a DRP, and that these two functions use segregated lumbar pathways. While this hypothesis cannot be excluded, the recent report by Karadimas and collaborators (*Karadimas et al., 2020*) strongly supports the hypothesis that the motor cortical command conveyed by the CST relays in rostral spinal segments. Indeed, they demonstrate that the cortical motor command to the hindlimb can be mediated by CS neurons projecting to the cervical cord, and then relayed by cervical-to-lumbar propriospinal neurons (*Karadimas et al., 2020*). Although we selectively study CS projecting at least to the lumbar cord (through lumbar retrograde infection), many CS neurons have collaterals at different spinal segments (*Kamiyama et al., 2015*). Therefore, their photostimulation at the sensorimotor cortex level would activate cervical pathways in addition to lumbar ones.

## Pathways conveying the cortical sensory control

Beyond inducing hindlimb muscle contraction, we have characterized the ability of the cortex to induce DRPs, which is a measure of PAD (*Wall and Lidierth, 1997*). We demonstrate that it is exclusively relayed through the CST and provide some information on spinal interneurons potentially mediating this effect. The most studied mechanism of segmental PAD involves a presynaptic GABAergic last order neuron (mostly expressing GAD65; *Betley et al., 2009*), located in laminae V–VI, under the control of a first-order excitatory interneuron that receives peripheral and descending (including cortical) inputs (*Eccles et al., 1962*; *Rudomin and Schmidt, 1999*; *Betley et al., 2009*). The targets of the CST identified by our transynaptic labeling are indeed predominantly located in laminae IV–VI and include a small population of GAD65-expressing neurons. They represent a potential substrate for cortically evoked DRP that would thus not require a spinal first-order excitatory interneuron.

In addition to the sensorimotor cortex, stimulation of other supraspinal nuclei is known to induce PAD. This is the case of the red nucleus and the reticular formation (*Jiménez et al., 1987*; *Sirois et al., 2013*), both of which also receive inputs from the sensorimotor cortex. The corticorubral and corticoreticular tracts were preserved by the pyramidotomy, yet the cortically induced PAD was completely lost. This strongly suggests that the ability of the red nucleus and reticular formation to induce PAD does not rely on a sensorimotor cortical command.

The control of incoming sensory information through PAD plays an important role in the generation of coordinated movements and processing of sensory information, including touch and nociceptive information {*Liu et al., 2018*, #2735; *McComas, 2016*, #2869}. GAD65-expressing neurons are essential for smoothed forelimb movements (*Fink et al., 2014*), but whether they require peripheral or cortical information to do so is yet unknown. Similarly, recordings in primates demonstrated that sensory information is inhibited in a modality- and phase-dependent manner during forelimb voluntary movements (*Seki et al., 2003*; *Seki et al., 2009*; *Seki and Fetz, 2012*), suggesting that this inhibition is somehow related to the motor command, but the exact underlying mechanism remains to be elucidated. Importantly, the change in afferent fiber excitability (*Wall, 1958*) accompanying the sensory presynaptic inhibition in primates was even smaller (<2 μV; *Seki et al., 2003*) than the DRPs we report as a consequence of cortical stimulation, suggesting that cortically evoked DRPs might be sufficient to significantly impact the transmission of sensory information. Here, we demonstrate that CS neurons from the same sensorimotor cortex area (but not necessarily a single population) are able to relay both motor command and sensory modulation to hindlimb, but through segregated pathways. Future studies on awake animals and/or focusing on specific sensory modalities are needed to further elucidate the timing and specificity of the sensory control we have described.

## Significance of the results

In contrast with the present results on hindlimbs, specific stimulation of the CST at the cervical level induces forelimb contraction (*Gu et al., 2017*), suggesting a direct relay of the cortical motor

command at this level (although antidromic activation of a collateral pathway cannot be excluded). Relative to the forelimb, mice do not display an extensive repertoire of skilled hindlimb movements and a difference in the level of cortical control is expected. The functional segregation we report here suggests that, while the sensory control is selectively targeted at the lumbar level, the hindlimb motor command is relayed in the upper cord. The fact that the upper cord (possibly the cervical cord according to *Karadimas et al., 2020*), receiving the motor command from the forelimb sensorimotor cortex, also receives the information of hindlimb motor command offers a substrate for the coordinated control of fore- and hindlimb, which is essential for most, if not all, skilled movements.

The organization of the CST greatly differs between species. From an anatomical point of view, this tract travels mainly in the dorsal funiculus in rodents, while it is located in the dorsolateral funiculus in primates, and its projection area within the spinal cord accordingly differs (*Lemon, 2008*). The evolution of the CST has been linked to the development of the most dexterous movements that can be performed by primates, with the appearance of direct cortico-motoneuronal (CMN) contacts that are absent in rodents. From an evolutionary perspective, it is interesting to note that the lumbar branch of the CST, which will later acquire a direct motor command role (including with direct CMN contacts in primates *Porter, 1985*), instead has predominant projections in the rodent dorsal horn for preferential modulation of afferent pathways. Importantly, primate CS neurons do not only form CMN contacts but also abundantly project onto spinal interneurons, and the sensory control function of the CST we report has also been proposed in primates (*Lemon, 2019*). While extrapolation of our results to primates is difficult in view of the large species differences, mice are the main and foremost animal model, including for pathologies affecting the CST such as spinal cord injury or amyotrophic lateral sclerosis. A better understanding of the differential roles of the mouse CST in the cervical and lumbar cord is thus an essential endeavor to properly interpret data from these models.

## Conclusion

Altogether, our results support a segregation of pathways involved in cortically evoked sensory modulation vs. motor control. The direct CST is able to induce motor contraction, independently of its supraspinal collaterals, through spinal targets possibly located at the cervico-thoracic level. However, we show that motor command is largely mediated by non-CST pathways, most likely cortico-rubral, or -reticular ones. On the other hand, the major and essential role of the lumbar rodent CST appears to be hindlimb sensory modulation at the primary afferent level through a population of lumbar target neurons whose activation is sufficient to produce DRPs. This forces reinterpretation of previous studies aiming at promoting CST function in a therapeutic perspective, in mouse injury or neurodegenerative models. While most of these studies have exclusively considered motor command aspects, our results provide a new perspective to analyze these efforts by considering the primary role of rodent lumbar CST, that is, sensory modulation.

## Materials and methods
### Animal models

This study was carried out in strict accordance with the national and international laws for laboratory animal welfare and experimentation and was approved in advance by the Ethics Committee of Strasbourg (CREMEAS; CEEA35; agreement number/reference protocol: APAFIS#12982-2017122217349941 v3). The following mice strains were used (adult males and females): *thy1-ChR2* (Cg-Tg(Thy1-COP4/EYFP)18Gfng/J, Jackson Laboratory stock no: 007612), TdTomato-Flex (*Gt(ROSA)26Sor^{tm14(CAG-tdTomato)Hze}*/J, Jackson Laboratory stock no: 007914, Ai14), *ChAT-EGFP* (*von Engelhardt et al., 2007*), and GAD65-GFP that stands for gad2-GFP (*López-Bendito et al., 2004*). All lines except *gad2-GFP* were back-crossed (at least 10 generations) on a CD1 background. *gad2-GFP* were on a C57Bl/6J background. The mice were housed in the Chronobiotron facility (UMS3415, CNRS, University of Strasbourg) in accordance with the European convention 2010/63/EU on the protection of animals used for scientific purposes.

## Electrophysiological recordings

### EMG recording

Ketamine/xylazine anesthesia was chosen to minimally affect muscular tone. After the initial anesthesia (150 mg/kg of ketamine, Imalgene 1000, and 10 mg/kg of xylazine, Rompun 20%), complementary doses of ketamine were given when the animal was showing signs of awakening (additional doses bridged by short exposure to isoflurane 1–1.5%; *Tennant et al., 2011*). Concentric EMG needle electrodes (Myoline Xp) were inserted in the TA muscle of the animals and a ground electrode was inserted in the tail. Responses were collected with a recording amplifier (IR183A, Cygnus Technology) in series with a second amplifier (Brownlee Precision, model440). The traces were filtered with a bandwidth of 0.1 Hz to 10 kHz, recorded with Spike2 software (version 8.00, CED, Cambridge, UK), and analyzed off line with Clampfit (pCLAMP, version 10.7) and home-made Python routines (WinPython 2.7.10, Python Software Foundation). EMG responses were recorded after cortical or spinal photostimulation (see below), and the response to three consecutive stimulations was averaged for analysis. The area under the curve of the rectified average trace was calculated in a window of 30 ms, either 40 ms before (baseline noise, N) or 10 ms after (signal, S) the onset of the stimulation. The EMG response was presented as the S/N ratio. In CST lesion experiments, all S/N ratios were expressed as a % of the control S/N (average of three responses before the lesion). The noise level (S/N of 1) was similarly expressed as a % of the above-defined control S/N.

### DRP recording

Under isofluorane (1.5–2%) anesthesia, the mice spinal cord was fixed with the spinal cord unit of a stereotaxic frame (Narishige Instruments). A laminectomy was performed to expose the surface of lumbar L3–L6 spinal cord segments. A dorsal root (L4–L6) was dissected, cut before the DRG, and a rootlet was suctioned with a glass pipet (sometime two rootlets were suctioned together). An agar pool was created on the exposed spinal cord and filled with NaCl 0.9% . The amplifiers/filters/software used were the same as for the EMG recordings. DRPs in response to 60 successive cortical or spinal photostimulations (see below) were averaged for analysis. The amplitude of the response from the onset to the peak was measured using Clampfit 10.0. The amplitude of the noise was similarly measured on a window preceding photostimulation. For spinal photostimulation, DRPs and EMGs recordings of three series of photostimulation were averaged.

For electrophysiological recordings, we excluded animals that had a constant decreasing signals (run-down) in otherwise stable conditions. For DRP recordings that required a delicate surgery, animals were excluded from the study if the surgical preparation/recovery was compromised or if scaring was abnormal after intraspinal AAV injection.

### Single-unit extracellular recording

The animal was prepared and anesthetized as for DRP recordings (except the dorsal root dissection). Single-unit extracellular recordings were made with a glass electrode (Harvard Apparatus, Holliston, MA, USA) filled with 0.5 M NaCl and 0.06 M HEPES, pH adjusted at 7.3 (resistance ~19 MΩ). A motorized micromanipulator (Narishige, Tokyo, Japan) was used to gradually insert the electrode with 4 µm steps. The amplifiers and software used were the same as for the DRP and EMG recordings, but the signal was at 0.1–3 kHz. The signal was analyzed with Python routines for spike detection (amplitude threshold of 0.30 mV) and delay measure (from onset of photostimulation to spike peak).

## Cortical photostimulation

Photostimulation of the cortex was performed on anesthetized animals while recording EMG or DRP. For the cortex, a craniotomy of approximately 9 mm² was performed under stereomicroscope on the contralateral side from the recording site. Muscle contraction, measured by EMG, was evoked by photostimulation with a 250 µm probe (473 nm laser source PSU-III-FDA, 56 mW/cm²) using the following protocol: trains of six pulses, 1 ms duration, 2 ms interval; 15 s minimum between each train (*Li and Waters, 1991*; *Carmel and Martin, 2014*). Cortically evoked DRPs typically require trains of electrical stimulations (*Andersen et al., 1962*; *Andersen et al., 1964*; *Eguibar et al., 1994*; *Wall and Lidierth, 1997*), a protocol that we adapted for photostimulation using a 105 µm probe (460 nm LED source, Prizmatix, 3.36 mW/cm²): trains of five pulses of 8 ms, 1 ms interval and 2 s between each train. In both cases, the probe was moved at the surface of the cortex along a 500-µm-spaced grid.

Isopotential contour plots were created, using for EMGs the S/N and for DRPs the peak amplitude (both normalized to the largest values in each animal) at each photostimulation point in the cortex, using a linear interpolation on a set of X, Y, Z triples of a matrix. For EMG, only S/N ratio above significance (Z-score >3, *Figure 2—figure supplement 2A*) was considered for the analysis. Then the isopotential contours were superimposed on the metric planes of the top view of the cortex (*Kirkcaldie et al., 2012*).

## Spinal cord photostimulation

Photostimulation of the spinal cord was performed on anesthetized mice after laminectomy (described above for DRPs recordings) using a 1.1 mm probe (LED source, 42 mW/cm²) first placed on the surface of the lumbar spinal cord (on top of the injection site). The stimulation trains were those used for cortical stimulation. If no signal was observed when stimulating on top of the injection site, the optical fiber was then swept rostrally and caudally on top the whole lumbar segment in order to try and obtain a signal. EMG and DRP recordings were sequentially performed and required to alternate between different types of anesthesia: isoflurane (craniotomy and laminectomy surgery), ketamine/xylazine (EMG recording), and isoflurane 1.5–2 % (DRPs recordings). Only the mice filling the two following criteria were kept for analysis: (1) TdTomato$^+$/EYFP$^+$ neurons were found in the stimulated area and (2) the response was not precisely associated to the onset or offset of optical pulses (which would be expected for a photoelectric artifact).

For spinal single-unit extracellular recordings, the photostimulation protocol consisted in either a single 1 ms pulse or in the same train used to test EMG responses (six pulses, 1 ms duration, 2 ms interval).

## Pyramidotomy

Electrolytic lesion of the CST was performed in a subset of Thy1-ChR2 and ChR2 retrogradely labeled mice, with 200 ms pulses, 30 mA using a constant current stimulator (Digitimer Ltd) through a silver bipolar electrode inserted in the pyramidal decussation (3.5–4 mm caudal to Bregma, 6 mm deep). The brain was removed for histological analysis at the end of the experiment.

## Cortical and spinal injections

Brain injections (1.5–2 mm lateral, 0.5–1 mm caudal to Bregma, 0.5 mm deep) were performed as previously described (*Cetin et al., 2006*) under isoflurane anesthesia (2–3%). Briefly, 90–270 nL of virus was injected by manual pressure using a 5 mL syringe coupled to a calibrated glass capillary under visual control. Spinal injections were performed using a similar manual pressure protocol. The pipette was inserted in the exposed space between two vertebrae (T13-L1, corresponding to spinal L4), as previously described (*Tappe et al., 2006*). 0.45 µL of virus was injected 300 µm lateral to the midline and 300–400 µm deep. In both cases, Manitol 12.5%  was injected i.p. (0.2–0.5 mL) after the surgery to enhance vector spread and improve transduction (*Tjølsen et al., 1992*). After spinal injection, transfected neurons were found on multiple segments centered in L4 (depending on the animals, up to the whole lumbar enlargement).

The animals were kept 2 weeks for the retroAAV injections and a minimum 5 weeks for dual injections before in vivo DRPs and/or EMGs recordings or histological analysis. Animals for which post-hoc histological analysis demonstrated inappropriate injections coordinates were excluded from the analysis.

## Viruses

AAV2/1-CBA-WGA-CRE-WPRE was purchased at the molecular tools platform at the Centre de recherche CERVO (Québec, Canada) and was used at a titer of $8 \times 10^{12}$ vg/mL. AAV-Ef1a-DIO ChETA-EYFP was a gift from Karl Deisseroth (*Gunaydin et al., 2010*; Addgene viral prep # 26968-AAV9; http://n2t.net/addgene:26968; RRID:Addgene_26968) and was used at a titer of $1 \times 10^{13}$ vg/mL. AAV-CAG-hChR2-H134R-tdTomato was a gift from Karel Svoboda (*Mao et al., 2011*; Addgene viral prep # 28017-AAVrg; http://n2t.net/addgene:28017; RRID:Addgene_28017) and was used at a titer of $7 \times 10^{12}$ vg/mL. pAAV-hSyn-EGFP was a gift from Bryan Roth (Addgene viral prep # 50465-AAVrg; http://n2t.net/addgene:50465; RRID:Addgene_50465) and was used at a titer of $1 \times 10^{13}$ vg/mL.

## Histology

Mice were transcardially perfused with PB followed by 4% paraformaldehyde (PFA) in PB 0.1 M, or, if histological analysis followed electrophysiological recordings, the brain and spinal cord were post-fixed overnight in PFA 4% in PB 0.1 M. Serial 50 µm brain (coronal or sagittal) and spinal (transverse or sagittal) sections were performed on a vibratome (Leica VT1000 S) and mounted using a DAPI staining mounting medium (Vectashield, Vector Laboratories).

## Image analysis

The extent of the cortical injection site was estimated by measuring the spread of reporter protein fluorescence (TdTomato) in the layer V of the cortex. The intensity of fluorescence was analyzed as previously described (*Lorenzo et al., 2008*; *Mesnage et al., 2011*) on evenly spaced (250 µm apart) transverse brain sections imaged with a microscope (Axio Imager 2, Zeiss). Briefly, the 'Plot Profile' function of ImageJ software (W. Rasband, National Institutes of Health) was used to measure the intensity of fluorescence along the horizontal axis of a 6000 × 250 µm rectangle containing the layer V of the cortex (*Figure 1—figure supplement 1*). These values were normalized to the highest intensity for each animal and used to plot a density map of each injection site. The contour of the injection area was defined as 30% of the maximal intensity, corresponding to the approximate intensity of individual neurons.

### Spatial distribution of CS postsynaptic neurons

Spinal lumbar sections (50 µm) were mounted serially and imaged using a Zeiss epifluorescence micro-scope or a Leica SP5 II confocal microscope. Images were aligned manually with Photoshop using the central canal and the dorsal funiculus as landmarks. Each labeled neuron was assigned coordinates corresponding to its distance to the center of the central canal (X and Y) and the index of the slice containing it (Z). This was used to calculate the number of labeled neurons every 100 µm × 100 µm bins (X × Y) (or 200 µm × 200 µm X × Z) in order to build density plots. The neuronal distribution was plotted using the 'spline 16' interpolation method of the matplotlib library in a homemade Python script.

## Quantification and statistical analysis

No prior sample size calculation was performed. The mean and standard deviation of the noise were measured from the three noise values obtained from each EMG recording and used to calculate the Z-scores. The Z-score of each EMG response to the stimulation was calculated using the following equation:

$$Z - \text{score} = \frac{\text{Signal- mean Noise}}{\text{standard deviation of Noise}}$$

A Z-score of 1.96 or 3 corresponding to a significance level of 0.05 or 0.001 was chosen to discriminate significant responses from nonsignificant ones. The conservative choice of threshold was chosen in agreement with visual inspection of the traces to ensure that any EMG responses was not simply due to noise. For each tests, the normality and variance equality were verified by SigmaPlot (version 13, 2014 Systat Software, Inc) using Shapiro–Wilk and Brown–Forsythe's methods, before we applied the parametric tests. All the data were analyzed using a one-way ANOVA and a post-hoc test was performed only if an effect showed statistical significance. The p-values for multiple comparison were measured using Holm–Sidak's method. Error bars in all figures represent mean ± SEM, *p<0.05, NS indicates no statistical significance (p≥0.05). The power of the tests was systematically attested to be above the desired power of 0.80.

## Acknowledgements

We gratefully acknowledge the support from the CNRS and Strasbourg University; University of Stras-bourg Institute for Advanced Study (USIAS) and ANR-13-JSV4-0003-01 grants to MCE; ANR-15-CE37-0001-01 CeMod, ANR-19-CE37-0007-03 MultiMod, and ANR-19-CE16-0019-01 NetOnTime to PI. YML was supported by a postdoctoral fellowships from the Fondation pour la Recherche Médicale (SPF20160936264) and CB was funded by a fellowship from the Ministère de la Recherche and by the

ARSLA (Association pour la Recherche sur la SLA). CB was a student of the EURIDOL graduate School of Pain (University of Strasbourg and ANR-17-EURE-0022). We also thank the following for providing/generating specific mouse lines: Gábor Szabó for GAD65-GFP, Jakob von Engelhardt for ChAT- EGFP. We thank Dr. Sophie Reibel-Foisset and the staff of the animal facility (Chronobiotron, UMS 3415 CNRS and Strasbourg University) for technical assistance. We also thank Yves De Koninck, Frédéric Doussau, and Didier Desaintjan for critical reading of the manuscript.

## Additional information

### Funding

| Funder | Grant reference number | Author |
|---|---|---|
| University of Strasbourg Institute for Advance Sciences | USIAS fellow | Matilde Cordero-Erausquin |
| Agence Nationale de la Recherche | ANR- 13-JSV4-0003-01 | Matilde Cordero-Erausquin |
| Agence Nationale de la Recherche | ANR - 19-CE37-0007-03 MultiMod | Philippe Isope |
| Agence Nationale de la Recherche | ANR - 19-CE16-0019-01 NetOnTime | Philippe Isope |
| Fondation pour la Recherche Médicale | Postdoctoral fellowship | Yunuen Moreno-Lopez |
| Ministère de l'Enseignement supérieur, de la Recherche et de l'Innovation | Graduate Student fellowship | Charlotte Bichara |
| Association pour la Recherche sur la Sclérose Latérale Amyotrophique et autres Maladies du Motoneurone | Graduate Student fellowship | Charlotte Bichara |
| EURIDOL graduate school | ANR-17-EURE-0022 | Charlotte Bichara |

The funders had no role in study design, data collection and interpretation, or the decision to submit the work for publication.

### Author contributions

Yunuen Moreno-Lopez, Conceptualization, Data curation, Formal analysis, Investigation, Methodology, Validation, Visualization, Writing – review and editing; Charlotte Bichara, Conceptualization, Data curation, Formal analysis, Investigation, Methodology, Software, Validation, Visualization, Writing – review and editing; Gilles Delbecq, Formal analysis, Investigation, Visualization; Philippe Isope, Conceptualization, Writing – review and editing, Funding acquisition, Methodology; Matilde Cordero-Erausquin, Conceptualization, Data curation, Formal analysis, Funding acquisition, Methodology, Project administration, Supervision, Validation, Visualization

### Author ORCIDs

Yunuen Moreno-Lopez (ID) http://orcid.org/0000-0002-4023-849X
Charlotte Bichara (ID) http://orcid.org/0000-0002-2154-3915
Philippe Isope (ID) http://orcid.org/0000-0002-0630-5935
Matilde Cordero-Erausquin (ID) http://orcid.org/0000-0002-1623-1773

### Ethics

This study was carried out in strict accordance with the national and international laws for laboratory animal welfare and experimentation and was approved in advance by the Ethics Committee of Strasbourg (CREMEAS; CEEA35; agreement number/reference protocol: APAFIS# 12982 - 2017122217349941 v3).

Decision letter and Author response
Decision letter https://doi.org/10.7554/eLife.65304.sa1
Author response https://doi.org/10.7554/eLife.65304.sa2

---

## Additional files

### Supplementary files
• Transparent reporting form

### Data availability
All raw data for each figure have been made available on Zenodo.

The following dataset was generated:

| Author(s) | Year | Dataset title | Dataset URL | Database and Identifier |
|---|---|---|---|---|
| Moreno-Lopez Y, Bichara C, Delbecq G, Isope P, Cordero-Erausquin M | 2021 | The corticospinal tract primarily modulates sensory inputs in the mouse lumbar cord (Raw data) | https://doi.org/10.5281/zenodo.5484040 | Zenodo, 10.5281/zenodo.5484040 |

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
