## [Decision Letter]

**Acceptance summary:**

This publication is of interest to scientist within the field of motor control and central modulation of sensory inputs, with original data likely to have a major impact on our understanding of the function of the corticospinal tract. By combining optogenetic stimulation with in vivo electrophysiology, it demonstrates that the corticospinal tracts' main direct action on the lumbar spinal cord is to modulate sensory transmission via primary afferent depolarization. Derived insights will help guide future studies on the organization of corticospinal modulation of sensory inputs.

**Decision letter after peer review:**

Thank you for submitting your article "Lumbar corticospinal tract in rodents modulates sensory inputs but does not convey motor command" for consideration by *eLife*. Your article has been reviewed by 3 peer reviewers, and the evaluation has been overseen by a Reviewing Editor and Ronald Calabrese as the Senior Editor. The following individuals involved in review of your submission have agreed to reveal their identity: David Bennett (Reviewer #1); Klas Kullander (Reviewer #2); Shawn Hochman (Reviewer #3).

Summary:

The study by Cordero-Erausquin investigates pathways for actions of the corticospinal tract involved in differentially regulating motor outputs and sensory inputs. It combines optogenetic stimulation within the sensorimotor cortex of specific corticospinal neurons based on the development of particular intersectional viral strategies with in vivo electrophysiology/ electromyography. They demonstrate that the modulation of sensory inputs from the limb and that of muscle contraction (motor output) are involving different corticospinal pathways. Thus, this paper argues in favor of reconsidering the functions of the corticospinal tract towards a possible additional prominent role in the modulation of sensory inputs. This study is of potential interest to scientists within the field of motor control interested in the organization of corticospinal modulation of sensory inputs.

Essential Revisions:

Three reviewers and I have carefully read the manuscript and came together to the conclusion that while this paper addresses important questions regarding the potential role of the corticospinal tract in regulating motor outputs and sensory inputs through modulation of sensory transmission via primary afferent depolarization (PAD), it contains major flows that must be corrected before being able to judge the suitability for publication in *eLife*. The three following major concerns must be addressed:

1 – The size of PAD evoked by the opto-stimulation of cortical tract is abnormally small. This could be due to the extracellular recording approach (bad seal of the extracellular electrode), to only a few neurons being transynaptically labelled or to the fact that the CST has negligible effects on PAD (that would be problematic for the interpretations and conclusions reached). Thus the authors must provide data allowing to discriminate between these different hypotheses to also support the conclusion made by the authors. In addition, a positive control measurement of PAD evoked by dorsal root stimulation is lacking. Indeed it would be important to calibrate the PAD size to a known afferent input like dorsal root stimulation.

2 – Anatomical and functional controls must be provided for the different viral infections and CRE-lines used in the study, to make sure that the proper populations are targeted and that this offers the possibility of having a very selective stimulation. The authors must provide rigorous controls of the tracing and optogenetics experiments.

3 – Some experimental series are done using a different anesthesia protocol (ketamine vs isoflurane). It remains possible that this might change the results of the experiments and explain the lack of EMG stimulation. Therefore experiments should be replicated using the same anesthetical agent.

*Reviewer #1:*

1) The size of the PAD evoked by corticospinal (CS) activation is exceedingly small, 0.5 to 1 uV (not mV), and required 30 – 60 trace averages to see PAD above the noise. This could be because the CST has negligible effects on PAD (which I doubt), or it could be that the recording suction electrodes do not provide a good seal on the dorsal roots and thus much of the signal is lost to extracellular leak currents in the root. This seal can be improved with grease or sucrose, though this is not necessary here (Stys et al. 1993; Huxley and Stampfli, 1951). Instead, at a minimum, it is imperative to compare the authors' CS evoked PAD to classical dorsal root evoked PAD. If they are similar in size then the authors enthusiasm about the CST controlling PAD holds.

2) The Thy-cre animal labels many cell in addition to lamina V CST neurons, and the authors should state this limitation when they introduce the model. This limitation is overcome by the very nice duel virus labelling ChETA insertion model (line 164), since in this case only the CST is labelled. However, the authors only test whether in this model light evokes EMG. It is critical that the authors also show us the light evoked PAD in this ChETA CST model. This will provide direct evidence for CST evoked PAD.

*Reviewer #2:*

Moreno-Lopez et al., investigated the contribution of the corticospinal tract in motor control and the spinal cord sensory gating. Using tracing tools and in vivo electrophysiology recordings, they mapped the cortical regions responsible for hindlimb muscle contractions and lumbar dorsal root potentials and showed that they are overlapping. They found that the branch of corticospinal tract targeting lumbar interneurons in the spinal cord is involved in mediating dorsal root potentials and by consequence plays a role in the regulation of sensory inputs. On the other hand, the corticospinal tract performs its function in hindlimb motor control via a relay in neurons of the upper spinal cord which then project to the lumbar spinal cord. The data of this manuscript are of great interest to go further in the understanding of the roles that play the corticospinal tract however the lack of rigorous controls of the tracing and optogenetics experiments impede the data to fully support the claims of the authors.

1 – The external stimulation of layer V pyramidal neurons of the sensorimotor cortex in Thy::ChR2 mice seems to elicit both DRPs and EMGs (Figure 1A and B). However, this experiment lacks essential controls to show that the DRP and muscle contraction are not just triggered by the photostimulation of the brain but is really due to the activation of the layer V pyramidal neurons.

The exact same recordings while illuminating the cortex in mice that are not expressing Chr2 is recommended.

It should also be stated on line 80, results, that these mice express ChR2 in other ares of the cortex and brain as well according to Arenkiel et al.

2 – Several statements in this publication rely on the use of the AAV-WGA-CRE tracing from the cortex to the spinal cord. The authors claim that this tracing is anterograde and monosynaptic. However, several publications report that WGA can be bidirectionally transported and can be transmitted polysynaptically (e.g. Levy et al. 2015 Neural Tracing Methods: Tracing Neurons and Their Connections). It seems that the direction of transportation of WGA and its ability to be transsynaptically transmitted depends on the neuronal network. Moreover, they use an AAV2/1 virus and the AAV1 serotype can also be transported in the neurons both anterogradely or retrogradely. Thus, it would be essential to show that the spinal cord traced cells result from the anterograde monosynaptic transmission of the WGA-Cre virus. This is also true for the experiments with ChAT::GFP, in what direction (anterograde, retrograde) do the authors envisage the tracing to take place? An inset in figure supplement 3 would be good to explain this better.

Recommendation: This could be performed using an IHC with an antibody targeting WGA in the spinal cord to prove that the Tomato cells express WGA. If there is any retrograde transport of the WGA-Cre, it might be possible to observe some staining in the afferent tracts in the spinal cord. I would suggest to add a representative picture of the whole spinal cord at the level of the traced neurons in addition to the crop picture in Fig1E. Perhaps a picture of a transverse or sagittal section from the midbrain and/or the brainstem where motor and sensory tracts are well distinct could also confirm that there is no staining in the sensory ascending tracts.

3 – There is a large disparity in the number of td-tom cells that were counted in GAD65::GFP and ChaT::GFP (427 neurons in GAD65::GFP from 3 mice and 45 in ChaT::gfp from 4 mice). Why such a disparity? Were there less cells in the ChaT::GFP tracing or did the authors just count less spinal cord sections?

It would be interesting to know the average number of Tomato neurons traced in the spinal cord and if this number is consistent between animals.

4 – The conclusion from Figure1 is that most of the spinal cord neurons targeted by CST are excitatory. Direct evidence that most of these neurons are really glutamatergic would reinforce the data. One possibility would be to perform in situ hybridization with a Vglut2 probe.

5 – There is a large difference in the intensity of the DRPs between Figure 1 (between 20 and 30uV) and Figure 2 (3.8uV). Why?

6 – The optogenetic stimulation performed in Figure 3 lacks a similar control as the one performed in Figure 1.

7 – In Figure 4. the spinal cord optogenetic experiments lack controls to check that the light by itself does not affect the recordings.

The neurons targeted by the optogenetic stimulation are located deep in the dorsal horn. Few in vivo spinal cord optogenetic stimulation experiments have been published so far. Thus, little evidence of the reliability of such experiments is available and a control showing that these neurons are indeed activated by the light stimulation would strengthen the data.

WGA-Cre was injected in the brain and is by consequence expressed as well in the corticospinal tract. Is there any ChETA-eYFP expression in the CST?

How many spinal cord neurons were ChETA-eYFP positive?

For this particular experiment, the authors had to switch between ketamine and isoflurane anesthesia. Could the lack of EMG stimulation be a consequence of this different anesthesia protocol?

Recommendation: To reproduce the spinal cord optogenetic stimulation experiment in animals that do not express Chr2 (ideally with a control virus injected). To check for the activation of the spinal cord neurons, for example, an immunohistochemistry against c-fos could be performed after light stimulation. Add a representative picture of the whole spinal cord to show that there is no expression of CHETA-eYFP in the corticospinal tract.

Further, to demonstrate that the particular anesthesia protocol used for this experiment is not the cause of the absence of muscle contraction, the same anesthesia protocol while performing the same optogenetic stimulation experiment shown in Figure 3 should be done (retro AAV-ChR2-mcherry in the spinal cord and light stimulation of the cortex).

*Reviewer #3:*

This study combines optogenetic recruitment of corticospinal tract neurons within mouse sensorimotor cortex, differentiated from other indirect corticofugal pathways via pyramidotomy, with various AAV-based transgene delivery approaches to compare and contrast corticospinal presumed GABAergic presynaptic inhibitory actions on unidentified hindlimb primary afferents in lumbar cord (measured as dorsal root potentials as an index of primary afferent depolarization) versus recruitment of tibialis anterior ankle flexor motor neurons (measured electromyographically).

The use of AAV cortical delivery of genetic construct generating WGA-Cre fusion protein in double transgenic Cre-dependent tdTomato/GAD65-GFP mice or in Cre-dependent tdTomato mice with additional intraspinal AAV Cre-dependent CHR2 variant (CHETA) delivery provided powerful molecular tools to demonstrate; (i) that corticospinal neurons projected to a small but significant fraction of GAD65 last-order inhibitory interneurons as well as many interposed putative excitatory interneurons involved in generating primary afferent depolarization (PAD), and (ii) that a distributed albeit small population of predominantly corticospinal tract (CST)-based transsynaptic optogenetically-labeled/recruited deep dorsal horn interneurons are capable of generating PAD. These are important and compelling observations.

The major conclusions are that sensorimotor cortical control of spinal hindlimb presynaptic afferent input via PAD is by corticospinal actions onto lumbar interneurons yet activation of a similar cortical region instead controls motor excitability via cortical projections to brainstem descending and/or more rostral spinal regions. However, one cannot exclude the possibility that differentiable actions may also arise due to use of different optogenetic cortical stimulation paradigms for studies on sensory input versus motor output systems. No explanation or rationale is given for the use of illumination area and stimulus train differences in studies on motor versus sensory function.

The anatomical/molecular studies are elegant and well described. It would have been helpful if anatomical studies included use of additional fluorescent immunolabeling To further identify transsynaptic WGA labeled interneurons as being excitatory (figure 1) or inhibitory (figure 4).

Overall, the work presents important and compelling observations on the organization of corticospinal projections onto interneurons in the lumbar spinal cord and distinct pathways involved in cortical control of sensory input versus motor output. Derived insights will help guide future studies on the organization of corticospinal modulation of sensory input.

The greatest weakness of the paper is the inability to study cortical recruitment of motor output (via EMG) and afferent input (via recorded DRPs in a dorsal root) simultaneously in the same animals under the same photostimulation (and anesthetic) protocols. There were differences in (i) photostimulation probes sizes (250 vs 105 μm), (ii) stimulus paradigms, (iii) anesthetics (ketamine vs isoflurane), and likely different optical power (laser vs LED) applied to assess lumbar spinal motor versus sensory function, respectively. How these variables bias neuronal in pathway recruitment was not considered. While use of different anesthetics may have been justifiably argued on ethical grounds, it unfortunately does not detract from the possibility that such differences may differentially alter neuronal recruitment, pathways and circuit excitability.

The work presents important and compelling observations on the organization of corticospinal projections onto interneurons in the lumbar spinal cord and distinct pathways involved in cortical control of sensory input versus motor output. Derived insights will help guide future studies on the organization of corticospinal modulation of sensory input.

This study combines optogenetic recruitment of corticospinal tract neurons within mouse sensorimotor cortex, differentiated from other indirect corticofugal pathways via pyramidotomy, with various AAV-based transgene delivery approaches to compare and contrast corticospinal presumed GABAergic presynaptic inhibitory actions on unidentified hindlimb primary afferents in lumbar cord (measured as dorsal root potentials as an index of primary afferent depolarization) versus recruitment of tibialis anterior ankle flexor motor neurons (measured electromyographically).

The use of AAV cortical delivery of genetic construct generating WGA-Cre fusion protein in double transgenic Cre-dependent tdTomato/GAD65-GFP mice or in Cre-dependent tdTomato mice with additional intraspinal AAV Cre-dependent CHR2 variant (CHETA) delivery provided powerful molecular tools to demonstrate; (i) that corticospinal neurons projected to a small but significant fraction of GAD65 last-order inhibitory interneurons as well as many interposed putative excitatory interneurons involved in generating primary afferent depolarization (PAD), and (ii) that a distributed albeit small population of predominantly corticospinal tract (CST)-based transsynaptic optogenetically-labeled/recruited deep dorsal horn interneurons are capable of generating PAD. These are important and compelling observations.

The major conclusions are that sensorimotor cortical control of spinal hindlimb presynaptic afferent input via PAD is by corticospinal actions onto lumbar interneurons yet activation of a similar cortical region instead controls motor excitability via cortical projections to brainstem descending and/or more rostral spinal regions. However, one cannot exclude the possibility that differentiable actions may also arise due to use of different optogenetic cortical stimulation paradigms for studies on sensory input versus motor output systems. No explanation or rationale is given for the use of illumination area and stimulus train differences in studies on motor versus sensory function.

The anatomical/molecular studies are elegant and well described. It would have been helpful if anatomical studies included use of additional fluorescent immunolabeling To further identify transsynaptic WGA labeled interneurons as being excitatory (figure 1) or inhibitory (figure 4).

The greatest weakness of the paper is the inability to study cortical recruitment of motor output (via EMG) and afferent input (via recorded DRPs in a dorsal root) simultaneously in the same animals under the same photostimulation (and anesthetic) protocols. There were differences in (i) photostimulation probes sizes (250 vs 105 μm), (ii) stimulus paradigms, (iii) anesthetics (ketamine vs isoflurane), and likely different optical power (laser vs LED) applied to assess lumbar spinal motor versus sensory function, respectively. How these variables bias neuronal in pathway recruitment was not considered. While justification for anesthetic differences may have been justifiably argued on ethical grounds, it unfortunately does not detract from the possibility that such differences may differentially alter neuronal recruitment, pathways and circuit excitability. It seems like experiments that record DRPs also recording EMG would be helpful to discriminate differences.

It may be helpful to undertake some simple experiments in Thy1::ChR2 mice to explore use of an anesthetic that permit simultaneous study of effects of cortical optogenetic stimulation on both sensory and motor systems in the same animal using identical photostimulation parameters, as well as for back-comparison to the different parameters used in the presently presented separate studies on corticospinal control of motor and sensory transmission. An anesthetic to consider is injectable long-lasting urethane anesthesia (Maggi and Meli 1986: 10.1007/bf01952426) which is only allowed for non-survival surgery that has been used in many studies on PAD (e.g. Lidierth 2006 – J Physiology) and likely has robust EMG activity (e.g. Zhang, C., et al., (2018) DOI: 10.5213/inj.1835052.52). While veterinary approval of urethane use is discouraged due to carcinogenic potential, use of protective clothing should easily justify its use to vetrinary staff given it is a very reliable and effective anesthetic with strong scientific rationale.

Although I fully appreciate the difficulty of the experiments, more insight on organization could have been provided using limited additional electrophysiological approaches for characterization of (i) motor and (ii) sensory pathways. (i) EMG recordings are rather easy in terminal experimentation, so it would have been helpful to know whether the cortical stimulation site chosen to recruit distal limb musculature (the foot flexor tibialis anterior) had somatotopic selectivity by also recording from a knee and hip flexor muscle. This could be incorporated into the additional urethane experiments suggested above. (ii) Use of an additional dorsal root or peripheral nerve for electrical stimulation in condition-test paradigms with CST photostimulation could have verified the expectation that corticospinal modulation of PAD was on interneurons interposed in low threshold afferent pathways (as characterized in the cat by Rudomin and colleagues) or when C-fibers as shown by this group in rat in one of their earlier studies (Moreno-López, Y., et al., (2013). PLoS One 8(7): e69063).

---

## [Author Response]

Essential revisionsThree reviewers and I have carefully read the manuscript and came together to the conclusion that while this paper addresses important questions regarding the potential role of the corticospinal tract in regulating motor outputs and sensory inputs through modulation of sensory transmission via primary afferent depolarization (PAD), it contains major flows that must be corrected before being able to judge the suitability for publication in eLife. The three following major concerns must be addressed:1 – The size of PAD evoked by the opto-stimulation of cortical tract is abnormally small. This could be due to the extracellular recording approach (bad seal of the extracellular electrode), to only a few neurons being transynaptically labelled or to the fact that the CST has negligible effects on PAD (that would be problematic for the interpretations and conclusions reached). Thus the authors must provide data allowing to discriminate between these different hypotheses to also support the conclusion made by the authors. In addition, a positive control measurement of PAD evoked by dorsal root stimulation is lacking. Indeed it would be important to calibrate the PAD size to a known afferent input like dorsal root stimulation.

We acknowledge the reviewers concerns about the size of the cortically-evoked dorsal root potentials (DRPs). We did some additional experiments presented below but we would like here to stress two important points. First, we are recording dorsal root potentials, not directly PAD. While PAD yields to deflections in the (low) mV range (recorded intra-axonally), DRPs are classically in the μV range, as observed originally in the Cat by Gossard and collaborators performing both these approaches (Wall, 1994). DRP recordings have rarely been performed in mice, and one of the few articles presenting them illustrate signals in the 100-150 μV range (Grunewald and Geis, 2014).

Second, as well noted by the reviewing editor above, the size of the DRP depends on the source eliciting it. The mouse study by Grunewald and Geis only presents segmentally-evoked DRPs. A seminal article by Wall and Lidierth (Wall and Lidierth, 1997) demonstrates that, in Rats, Cx-evoked DRPs are approximatively of 25μV amplitude, while segmental DRP are much larger, several hundreds of μV. Large segmental DRPs are also observed in the original publication by the same author (Wall, 1994).

We have now provided a direct measure of the size of the DRP evoked by the opto-stimulation of the cortical tract, independent of the transynaptic experiment. We have performed DRP recordings in mice where CS neurons are retrogradelly infected by a ChR2-encoding virus (Figure 3 —figure supplement 1). We also present in this figure the individual DRP amplitudes for all the recordings performed in the study, i.e. in three different mouse models, means presented in Author response table 1.

**Author response table 1. sa2table1:** 

Mouse model	Stimulation site	N	Mean DRP ampl.	Range DRP ampl.
Thy1 mice	Cortex	10	7.0 μV	3.3 μV – 20.4 μV
ChR2 in CS targets	Spinal Cord	8	2.9 μV	0.9 μV – 5.0μV
ChR2 in CS neurons	Cortex	5	14.5 μV	8.4 μV – 26.8 μV

The DRPs amplitude recorded when the CS is exclusively stimulated is in the same range as the one recorded when the sensorimotor cortex is stimulated, and to similar study performed in Rats (Wall and Lidierth, 1997). It is indeed smaller when only the targets of the CS are stimulated in the spinal cord, as this approach is not expected to stimulate all of the targets of the CST. We did however try to perform segmental PAD recordings, but we were limited by our equipment that prevented the proper placement of the required electrodes in such a small available space in mice (in addition to the spinal clamps).

A last point we would like to mention is that a small deflection in the μV range does not mean that the physiological effect is negligible. Indeed, recordings in the monkey by Seki and collaborators has shown effective sensory presynaptic inhibition correlated with a modest (<2μV) change in the amplitude of the antidromic response recorded in the afferent nerve (Seki et al., 2003). This is now included in the discussion (l. 345-349).

2 – Anatomical and functional controls must be provided for the different viral infections and CRE-lines used in the study, to make sure that the proper populations are targeted and that this offers the possibility of having a very selective stimulation. The authors must provide rigorous controls of the tracing and optogenetics experiments.

We agree that the anatomical and functional controls presented in the original version of the manuscript were incomplete and we now provide additional ones in supplementary figures. These include :

– Lack of labelling in the spinal cord ascending tract or in the cerebellum deep nuclei after AAV WGA-Cre injection in the cortex, demonstrating that spinal TdTomato neurons have been labeled through transynaptic anterograde tracing (Figure 1 —figure supplement 2).

– Appropriate dorsal horn view to show that the CST does not target dorsal horn cholinergic interneurons (Figure 1 —figure supplement 3).

– DRP recordings after selective stimulation of CS neurons (after their retrograde infection by spinal injection of rAAV ChR2) (Figure 3 —figure supplement 1).

– Absence of EMG and DRP signal after cortical stimulation of GFP-expressing CS neurons (Figure 3 —figure supplement 1).

– Absence of correlation between the number of CS targets transynaptically labeled and the survival time or the cortical area infected (Figure 4 —figure supplement 1).

– Absence of labelling of the CST after the intersectional approach to induce ChETA expression in the lumbar CS targets (Figure 4 —figure supplement 2).

– Labeling of ventral horn neurons (corresponding to the deepest targets of the CS) with the intraspinal AAV ChETA-floxed injection (Figure 4 —figure supplement 2).

– Analysis of the light penetration after surface illumination of the dorsal horn: this photostimulation induced muscle contraction in Thy1::ChR2 animals, and directly activates neurons located as deep as 600 μm (single unit recordings) (Figure 4 —figure supplement 3).

3 – Some experimental series are done using a different anesthesia protocol (ketamine vs isoflurane). It remains possible that this might change the results of the experiments and explain the lack of EMG stimulation. Therefore experiments should be replicated using the same anesthetical agent.

We aimed at performing the two types of recordings (EMG and DRP) with the same anesthetical agent, unfortunately we do not believe it is possible (see below) and, most importantly, we do not believe it biases the result of the present study nor explains the lack of EMG signal after spinal photostimulation.

We are thankful to the reviewers for their suggestions to try and identify unifying recording conditions. Unfortunately recording cortically-evoked EMGs requires even more stringent conditions than spinally evoked-ones. For example, we now show (Figure 4 —figure supplement 3) that surface stimulation of the spinal cord induces EMG signal in Thy1::ChR2 animals under isoflurane anesthesia, but this was never the case after cortical stimulation. Similarly, we were not able to obtain Cx-EMG under urethane anesthesia in preliminary experiments. Ketamine, anesthesia that was the only condition where we obtained Cx-EMG; but this anesthesia was too light to be compatible with the large laminectomy required for DRP recordings (procedure refused by our ethical committee).

Importantly, we do not compare responses thresholds nor the extent of the cortical area producing a response in between types of recordings (EMG vs. DRP). Throughout the manuscript, we base our conclusions on comparisons of a given signal (EMG or DRP), with a given experimental approach, in different animal models (THY1, ChR2-retro, pyramidotomy, transynaptically labeled CS targets). These conclusions are therefore not impacted by the differences in recording conditions between EMGs and DRPs.

However, we have now expanded the “reliability of circuit investigations” section to further discuss the consequences of obtaining results with different anesthesia (and stimulation conditions in general).

Lastly, concerning the lack of EMG signal after local stimulation of the CST targets, we now provide additional arguments to rule out experimental biases. As explained below in the individual responses, we have previously perform a similar switch of anesthesia without impact on EMG responses. We also provide anatomical and functional controls demonstrating that our intraspinal AAV injections can reach the deepest CS targets and that surface illumination of the cord also penetrates enough to activate them.

Reviewer #1:1) The size of the PAD evoked by corticospinal (CS) activation is exceedingly small, 0.5 to 1 uV (not mV), and required 30 – 60 trace averages to see PAD above the noise. This could be because the CST has negligible effects on PAD (which I doubt), or it could be that the recording suction electrodes do not provide a good seal on the dorsal roots and thus much of the signal is lost to extracellular leak currents in the root. This seal can be improved with grease or sucrose, though this is not necessary here (Stys et al. 1993; Huxley and Stampfli, 1951). Instead, at a minimum, it is imperative to compare the authors' CS evoked PAD to classical dorsal root evoked PAD. If they are similar in size then the authors enthusiasm about the CST controlling PAD holds.

This point was raised by the reviewing editor and our response is to be found in the first page of this document.

2) The Thy-cre animal labels many cell in addition to lamina V CST neurons, and the authors should state this limitation when they introduce the model. This limitation is overcome by the very nice duel virus labelling ChETA insertion model (line 164), since in this case only the CST is labelled.

The expression in other cells is now stated when presenting this mouse line (l. 99-100). The limitations are indeed presented when introducing the retrograde virus model.

However, the authors only test whether in this model light evokes EMG. It is critical that the authors also show us the light evoked PAD in this ChETA CST model. This will provide direct evidence for CST evoked PAD.

We have now included DRP measurements after exclusive stimulation of CS neurons (from retrograde viral infection of a ChR2 encoding virus in the spinal cord). Stimulation of CS neurons indeed induced DRPs and this is now presented in supplementary Figure 3 —figure supplement 1.

Reviewer #2:1 – The external stimulation of layer V pyramidal neurons of the sensorimotor cortex in Thy::ChR2 mice seems to elicit both DRPs and EMGs (Figure 1A and B). However, this experiment lacks essential controls to show that the DRP and muscle contraction are not just triggered by the photostimulation of the brain but is really due to the activation of the layer V pyramidal neurons.The exact same recordings while illuminating the cortex in mice that are not expressing Chr2 is recommended.

We now include in supplementary Figure 3—figure supplement 1 the recordings obtained from animals injected with an EGFP-encoding retro-AAV in the spinal cord, where photostimulation of the cortex leads to no DRP or EMG signal. This control is now mentioned in l. 104-105 and 171-173.

It should also be stated on line 80, results, that these mice express ChR2 in other ares of the cortex and brain as well according to Arenkiel et al.

This is now indicated.

2 – Several statements in this publication rely on the use of the AAV-WGA-CRE tracing from the cortex to the spinal cord. The authors claim that this tracing is anterograde and monosynaptic. However, several publications report that WGA can be bidirectionally transported and can be transmitted polysynaptically (e.g. Levy et al., 2015 Neural Tracing Methods: Tracing Neurons and Their Connections). It seems that the direction of transportation of WGA and its ability to be transsynaptically transmitted depends on the neuronal network.

The tracing properties of WGA seem to depend on the mode of injection/expression. When the lectin itself is injected, it can indeed travel in the retrograde direction (LeVay and Voigt, 1990). However the seminal studies using WGA-expressing transgenic mice already mention an exclusive anterograde transneuronal tracing (Braz et al., 2002). This was later confirmed by several other groups, including for the specific WGA-Cre fusion AAV construct (Gradinaru et al., 2010; Libbrecht et al., 2017). One puzzling exception is the study by Xu and Südhof (Xu and Sudhof, 2013) where WGA-Cre (encoded in AAV) is presented as a transynaptic retrograde tracer. In many cases, regions are reciprocally connected and it is difficult to conclude unequivocally on the direction of the transfer. In Suppl. Figure 4 of the Xu and Südhof paper, for example, they show axons of primarily infected neurons (N. Reuniens neurons) in the cortex, next to transynaptically labelled cortical neurons. One could interpret that the cortical neurons have been labelled through transynaptic anterograde passage of the WGA-Cre from the WGA-Cre expressing axons in their very close vicinity; yet the authors conclude that the cortical neurons received WGA-Cre by retrograde tracing of cortico-reuniens neurons.

Concerning our study, the important point is to be sure that spinal labeled neurons are indeed direct targets of the cortex. As suggested by the reviewer below, we can exclude a retrograde tracing because spinal ascending tracts are devoid of labelling (Figure 1 —figure supplement 2). In the same figure, we also illustrate deep cerebellar nuclei, that are, like the spinal cord, two synapses away in the retrograde direction, and that show no labelling.

More generally, in contrast with a transynaptic virus that can be amplified in the target neuron (and then cross another synapse), it is the WGA-Cre fusion protein that is transported transneuronally and thus is highly diluted. For another study, we have used an AAV encoding for the WGA-EGFP fusion protein: even with long survival times, the EGFP was never visible in the target neurons without immunohistochemical amplification, suggesting low concentration after transynaptic transfer. Using the WGA-Cre constructs (in TdTomato-floxed mice) ensures that the synaptic targets can be identified without further amplification. However we, and the above cited studies, see no evidence of multiple synapse crossing. Some elements of this response are now included in Figure 1 —figure supplement 2.

Moreover, they use an AAV2/1 virus and the AAV1 serotype can also be transported in the neurons both anterogradely or retrogradely. Thus, it would be essential to show that the spinal cord traced cells result from the anterograde monosynaptic transmission of the WGA-Cre virus.

The WGA-Cre AAV1 virus is injected in the cortex. To explain the presence of Cre-expressing neurons in the spinal cord as the consequence of retrograde transport, we should hypothesize a retrograde transynaptic labeling, as there is no direct spino-cortical tract. As stated in the above response, we now include in Figure 1 —figure supplement 2 several elements demonstrating that the spinal labeling does not result from transynaptic retrograde transport.

This is also true for the experiments with ChAT::GFP, in what direction (anterograde, retrograde) do the authors envisage the tracing to take place? An inset in figure supplement 3 would be good to explain this better.

We thank the reviewer for this suggestion and have now included an inset in the figure to illustrate the anterograde transynaptic labeling in ChAT::EGFP mice. The demonstration of anterograde transport is the same as above as it is the same tool.

Recommendation: This could be performed using an IHC with an antibody targeting WGA in the spinal cord to prove that the Tomato cells express WGA. If there is any retrograde transport of the WGA-Cre, it might be possible to observe some staining in the afferent tracts in the spinal cord. I would suggest to add a representative picture of the whole spinal cord at the level of the traced neurons in addition to the crop picture in Fig1E. Perhaps a picture of a transverse or sagittal section from the midbrain and/or the brainstem where motor and sensory tracts are well distinct could also confirm that there is no staining in the sensory ascending tracts.

We are thankful for this suggestion that elegantly rules out retrograde labeling at the level of the spinal cord (which, again, does not project directly to the somatosensory cortex). We have now included a larger view of the spinal cord in Figure 1 —figure supplement 2, showing that labeling is restricted to the ventral part of the dorsal funiculus (where the dorsal corticospinal tract resides) and in the grey matter.

3 – There is a large disparity in the number of td-tom cells that were counted in GAD65::GFP and ChaT::GFP (427 neurons in GAD65::GFP from 3 mice and 45 in ChaT::gfp from 4 mice). Why such a disparity? Were there less cells in the ChaT::GFP tracing or did the authors just count less spinal cord sections?It would be interesting to know the average number of Tomato neurons traced in the spinal cord and if this number is consistent between animals.

We did a systematic counting in all animals and numbers are now included in Figure 4 – Supplementary Figure 1. There was a large variation in the numbers of transynaptic labeled neurons that was not correlated to survival time post-injection, nor to the size of the infected cortical area (this is now detailed in the suppl. Figure).

4 – The conclusion from Figure1 is that most of the spinal cord neurons targeted by CST are excitatory. Direct evidence that most of these neurons are really glutamatergic would reinforce the data. One possibility would be to perform in situ hybridization with a Vglut2 probe.

This conclusion required more literature support, as also pointed out by reviewer 1. Although we agree that a better characterization of the neurochemical nature of CS targets would be very interesting, the proposed in situ hybridization (on top of the transynaptic tracing) was beyond our present expertise. We thus took out this sentence as the important conclusion of this experiment is that some GAD65 neurons are directly targeted by the CST.

5 – There is a large difference in the intensity of the DRPs between Figure 1 (between 20 and 30uV) and Figure 2 (3.8uV). Why?

In addition to the plot now included in Figure 3—figure supplement 1, presenting DRP amplitudes, we present in Author response image 1 an additional dissection of the results depending on the type of experiment and on the experimenter. It so happens that the few animals on which the pyramidotomy was performed (n=3, THY1 prePYR in Author response image 1, and also in Figure 2) are among the THY1 animals where the smallest DRP amplitudes were recorded. The DRP amplitude does not depend so much on the experimenter, but rather on the quality of the dissection and size of the rootlet.

**Author response image 1. sa2fig1:** DRP amplitude as a function of experiment type and experimenter. THY1: experiments on THY1-ChR2 animals (Fig. 1), THY1 prePYT: experiments on THY1-ChR2 where a pyramidotomy was performed (Fig. 2), measures taken before the pyramidotomy, Retro: animals injected with ChR2-retro AAV in the spinal cord (Fig 3-Figure supplement 1), N+1: animals with a double viral injection inducing ChETA expression in the CS lumbar targets (Fig. 4). The first three are responses to cortical photostimulations, the last one corresponds to responses to spinal photostimulations.YML, CB, GD: the three experimenters that performed the recordings

Also, the group of animals where ChR2 in expressed in CS targets (named “N+1” in Author response image 1) repeatedly demonstrated a lower amplitude of DRPs. This was expected, as the efficacy of transynaptic transfer was limited; thus, only a fraction of the total CS targets could be activated.

6 – The optogenetic stimulation performed in Figure 3 lacks a similar control as the one performed in Figure 1.

This control is now included in suppl. Figure 3 —figure supplement 1.

7 – In Figure 4. the spinal cord optogenetic experiments lack controls to check that the light by itself does not affect the recordings.

Photostimulation of the spinal cord induced induced DRPs in the ipsilateral lumbar root of 8 out of 9 mice. In one mouse, the same photostimulation induced no significant DRP demonstrating that the light itself does not produce an artefact that we would misinterpret as a response.

The neurons targeted by the optogenetic stimulation are located deep in the dorsal horn. Few in vivo spinal cord optogenetic stimulation experiments have been published so far. Thus, little evidence of the reliability of such experiments is available and a control showing that these neurons are indeed activated by the light stimulation would strengthen the data.

We already mentioned in the discussion (l. 265) that we were able to record an intraspinal LFP as deep as 1150 μm after photostimulating the surface of the cord, in the double infected (ChETA in the CS targets) mice.

To directly address this issue with our experimental configuration, we performed additional controls in THY1 animals, now included in a new Figure 4 —figure supplement 3 (and described in l. 208-212). These experiments illustrate that surface illumination induces muscle contraction (and EMG signal, even under isoflurane anesthesia) as well as direct activation of ChR2-expressing ventral horn neurons. This demonstrates that the light efficiently penetrates the cord as deep as 600μm after surface illumination.

Finally, for spinal photostimulations we used our largest probe (1,1 mm) at a power of 42 mW/cm2. With powers equal or lower than this, and a similar surface illumation, Caggiano and colleagues were able to induce motor contraction in THY1 and ChAT-ChR2 animals (Caggiano et al., 2016); in the latter experiment, they consider stimulating directly motoneurons located in the deep ventral horn.

WGA-Cre was injected in the brain and is by consequence expressed as well in the corticospinal tract. Is there any ChETA-eYFP expression in the CST?

We now include a picture (Figure 4 —figure supplement 2) illustrating the absence of ChETA-eYFP expression in the CST. This demonstrates that the intraspinal injection of AAV1-Flex-ChETA does not lead to retrograde infection of CS neurons, and that only the lumbar targets of the CST express ChETA in this intersectional experiment. This is described in the main text on l. 204-206.

How many spinal cord neurons were ChETA-eYFP positive?

The numbers are now indicated in Figure 4 —figure supplement Figure 2. As stated there, these might be underestimated as the histological analysis was performed after a long recording session, with only post-fixation (no intracardiac perfusion of fixative).

For this particular experiment, the authors had to switch between ketamine and isoflurane anesthesia. Could the lack of EMG stimulation be a consequence of this different anesthesia protocol?

In another project, we routinely perform isoflurane to ketamine/xylazine switches of anesthesia, without compromising EMG responses, we thus do not believe that this might be an explanation for the lack of EMG response.

Recommendation: To reproduce the spinal cord optogenetic stimulation experiment in animals that do not express Chr2 (ideally with a control virus injected). To check for the activation of the spinal cord neurons, for example, an immunohistochemistry against c-fos could be performed after light stimulation.

We are not sure about the point the reviewer is willing to raise here. We now show a control (single unit recording) of a ventral horn neuron activated by the surface illumination. Is this answering the reviewer’s point? We also mentioned earlier that we did not observe DRP signal in 1 out of 9 mice, demonstrating that the signal we interpret as DRP is not produced by the light alone.

Add a representative picture of the whole spinal cord to show that there is no expression of CHETA-eYFP in the corticospinal tract.

We now include a picture of the whole spinal cord (Figure 4 —figure supplement 2) illustrating the absence of ChETA-eYFP expression in the CST.

Further, to demonstrate that the particular anesthesia protocol used for this experiment is not the cause of the absence of muscle contraction, the same anesthesia protocol while performing the same optogenetic stimulation experiment shown in Figure 3 should be done (retro AAV-ChR2-mcherry in the spinal cord and light stimulation of the cortex).

EMGs are always recorded in ketamine/xylazine. In this particular experiment (ChETA in the targets of the CST), we switch between isoflurane and ketamine/xylazine, but as mentioned above, this is a protocol we have used in another project without negatively impacting EMG measures.

Reviewer #3:The major conclusions are that sensorimotor cortical control of spinal hindlimb presynaptic afferent input via PAD is by corticospinal actions onto lumbar interneurons yet activation of a similar cortical region instead controls motor excitability via cortical projections to brainstem descending and/or more rostral spinal regions. However, one cannot exclude the possibility that differentiable actions may also arise due to use of different optogenetic cortical stimulation paradigms for studies on sensory input versus motor output systems. No explanation or rationale is given for the use of illumination area and stimulus train differences in studies on motor versus sensory function.

The two types of study involve stimulation of the same area: either mapping of the sensorimotor cortex for Figure 1, or a stimulation located at the center of the two maps for the following cortical stimulations. We have now included a reference to the articles that inspired our stimulation protocols (Methods l. 456-458).

The anatomical/molecular studies are elegant and well described. It would have been helpful if anatomical studies included use of additional fluorescent immunolabeling To further identify transsynaptic WGA labeled interneurons as being excitatory (figure 1) or inhibitory (figure 4).

Although we agree that a better characterization of the neurochemical nature of CS targets would be very interesting, reliable antibodies labeling large subpopulations of dorsal horn neurons are scarce, and most studies rely on the use of transgenic reporter mouse to identify, for example, excitatory neurons. We could test GAD65 mice which were important in our project as GAD65+ neurons are known to form presynaptic contacts onto proprioceptive terminals; we could thus demonstrate that some GAD65 neurons were directly targeted by the CST. Further characterization would require new tools that are beyond the scope of the present study.

The greatest weakness of the paper is the inability to study cortical recruitment of motor output (via EMG) and afferent input (via recorded DRPs in a dorsal root) simultaneously in the same animals under the same photostimulation (and anesthetic) protocols. There were differences in (i) photostimulation probes sizes (250 vs 105 μm), (ii) stimulus paradigms, (iii) anesthetics (ketamine vs isoflurane), and likely different optical power (laser vs LED) applied to assess lumbar spinal motor versus sensory function, respectively. How these variables bias neuronal in pathway recruitment was not considered.

We agree with the reviewer that many parameters differentiate the two types of recordings, and some of the consequences were already discussed in the “reliability of circuit investigations” section. We have now developed this section with a discussion on the recruitment threshold (l. 243-247). We do not believe that this will bias our results as we do not compare thresholds in between stimulation/recording configurations, but rather analyze the amplitude of a given signal (EMG or DRP), with a given experimental approach, in different animal models (THY1, ChR2-retro, pyramidotomy, transynaptically labeled CS targets).

While use of different anesthetics may have been justifiably argued on ethical grounds, it unfortunately does not detract from the possibility that such differences may differentially alter neuronal recruitment, pathways and circuit excitability.It may be helpful to undertake some simple experiments in Thy1::ChR2 mice to explore use of an anesthetic that permit simultaneous study of effects of cortical optogenetic stimulation on both sensory and motor systems in the same animal using identical photostimulation parameters, as well as for back-comparison to the different parameters used in the presently presented separate studies on corticospinal control of motor and sensory transmission. An anesthetic to consider is injectable long-lasting urethane anesthesia (Maggi and Meli 1986: 10.1007/bf01952426) which is only allowed for non-survival surgery that has been used in many studies on PAD (e.g. Lidierth 2006 – J Physiology) and likely has robust EMG activity (e.g. Zhang, C., et al., (2018) DOI: 10.5213/inj.1835052.52). While veterinary approval of urethane use is discouraged due to carcinogenic potential, use of protective clothing should easily justify its use to vetrinary staff given it is a very reliable and effective anesthetic with strong scientific rationale.

Obtaining an EMG response after cortical stimulation requires a specific and very light anesthesia. In the Zhang et al., paper, the EMGs obtained in urethane-anesthetized mice were not evoked by supraspinal stimulations. We have similarly obtained EMG signals after spinal stimulation (in THY1 animals) under light isoflurane stimulation (now illustrated in Figure 4 —figure supplement 3). However, we have unsuccessfully attempted to evoke EMG from cortical stimulation under urethane anesthesia in initial exploratory experiments.

Although I fully appreciate the difficulty of the experiments, more insight on organization could have been provided using limited additional electrophysiological approaches for characterization of (i) motor and (ii) sensory pathways. (i) EMG recordings are rather easy in terminal experimentation, so it would have been helpful to know whether the cortical stimulation site chosen to recruit distal limb musculature (the foot flexor tibialis anterior) had somatotopic selectivity by also recording from a knee and hip flexor muscle. This could be incorporated into the additional urethane experiments suggested above.

Although EMG recordings per se are not a difficult experimentation, obtaining a somatotopic map is very demanding because there is only a small window of time during which stimulations/recordings can be performed: the ketamine/xylazine anesthesia has to be light enough to enable movements, but without risking awakening of the animal. In practice, the experimenter had only a few seconds before an additional anesthesia dose was required, that just allowed to obtain the 3-4 replicates needed for a specific coordinate. Building the cortical map thus required multiple re-injection of anesthetics and we observed some tolerance that limited the duration of the experiment.

Our results perfectly coincided with studies published earlier by (Tennant et al., 2011). As they describe the full somatotopy of the hindlimb sensorimotor cortex, and because of the aforementioned difficulty in obtaining a map, we did not attempt to further reproduce these data.

(ii) Use of an additional dorsal root or peripheral nerve for electrical stimulation in condition-test paradigms with CST photostimulation could have verified the expectation that corticospinal modulation of PAD was on interneurons interposed in low threshold afferent pathways (as characterized in the cat by Rudomin and colleagues) or when C-fibers as shown by this group in rat in one of their earlier studies (Moreno-López, Y., et al., (2013). PLoS One 8(7): e69063).

We agree that elucidating the type of sensory modality modulated by the CS tract would be an important advance in the field, however at this stage we consider this as an interesting follow-up project requiring the development of specific expertise and technical approaches for it to be addressed in mice. It is mentioned in the perspectives of our discussion (l. 353-355).

References

Braz JM, Rico B, Basbaum AI (2002) Transneuronal tracing of diverse CNS circuits by Cre-mediated induction of wheat germ agglutinin in transgenic mice. Proc Natl Acad Sci U S A 99:15148-15153.

Caggiano V, Cheung VC, Bizzi E (2016) An Optogenetic Demonstration of Motor Modularity in the Mammalian Spinal Cord. Sci Rep 6:35185.

Cordero-Erausquin M, Inquimbert P, Schlichter R, Hugel S (2016) Neuronal networks and nociceptive processing in the dorsal horn of the spinal cord. Neuroscience:S0306-4522(0316)30421-30423.

Cui L, Kim YR, Kim HY, Lee SC, Shin HS, Szabo G, Erdelyi F, Kim J, Kim SJ (2011) Modulation of synaptic transmission from primary afferents to spinal substantia gelatinosa neurons by group III mGluRs in GAD65-EGFP transgenic mice. J Neurophysiol 105:1102-1111.

Gradinaru V, Zhang F, Ramakrishnan C, Mattis J, Prakash R, Diester I, Goshen I, Thompson KR, Deisseroth K (2010) Molecular and cellular approaches for diversifying and extending optogenetics. Cell 141:154-165.

Grunewald B, Geis C (2014) Measuring spinal presynaptic inhibition in mice by dorsal root potential recording in vivo. J Vis Exp.

LeVay S, Voigt T (1990) Retrograde transneuronal transport of wheat-germ agglutinin to the retina from visual cortex in the cat. Exp Brain Res 82:77-81.

Libbrecht S, Van den Haute C, Malinouskaya L, Gijsbers R, Baekelandt V (2017) Evaluation of WGA-Cre-dependent topological transgene expression in the rodent brain. Brain Struct Funct 222:717-733.

Seki K, Perlmutter SI, Fetz EE (2003) Sensory input to primate spinal cord is presynaptically inhibited during voluntary movement. Nat Neurosci 6:1309-1316.

Tennant KA, Adkins DL, Donlan NA, Asay AL, Thomas N, Kleim JA, Jones TA (2011) The organization of the forelimb representation of the C57BL/6 mouse motor cortex as defined by intracortical microstimulation and cytoarchitecture. Cereb Cortex 21:865-876.

Ueno M, Nakamura Y, Li J, Gu Z, Niehaus J, Maezawa M, Crone SA, Goulding M, Baccei ML, Yoshida Y (2018) Corticospinal Circuits from the Sensory and Motor Cortices Differentially Regulate Skilled Movements through Distinct Spinal Interneurons. Cell Rep 23:1286-1300 e1287.

Wall PD (1994) Control of impulse conduction in long range branches of afferents by increases and decreases of primary afferent depolarization in the rat. Eur J Neurosci 6:1136-1142.

Wall PD, Lidierth M (1997) Five sources of a dorsal root potential: their interactions and origins in the superficial dorsal horn. J Neurophysiol 78:860-871.

Xu W, Sudhof TC (2013) A neural circuit for memory specificity and generalization. Science 339:1290-1295.